# Interpreting mammalian synonymous site conservation in light of the unwanted transcript hypothesis

Matthew J. Christmas [1,2] ✉, Michael X. Dong[1,2], Jennifer R. S. Meadows [1,2], Sergey V. Kozyrev [1,2] & Kerstin Lindblad-Toh [1,2,3]

Mammalian genomes are biased towards GC bases at third codon positions, likely due to a GC-biased ancestral genome and the selectively neutral recombination-related process of GC-biased gene conversion. The unwanted transcript hypothesis posits that this high GC content at synonymous sites may be beneficial for protecting against spurious transcripts, particularly in species with low effective population sizes. Utilising a 240 placental mammal genome alignment and single-base resolution conservation scores, we interpret sequence conservation at mammalian four-fold degenerate sites in this context and find evidence in support of the unwanted transcript hypothesis, including a strong GC bias, high conservation at sites relating to exon splicing, less human genetic variation at conserved four-fold degenerate sites, and conservation of sites important for epigenetic regulation of developmental genes. Additionally, we show that high conservation of four-fold degenerate sites in essential developmental genes, including homeobox genes, likely relates to the low mutation rates experienced by these genes.

Evolutionary constraint, or limited sequence evolution over time due to strong purifying selection, is evident in functional regions of genomes. In protein-coding genes, the constraint is largely due to purifying selection acting on mutations that alter the amino acid sequence and disrupt protein function, while mutations at synonymous sites that have no effect on the amino acid specified were originally considered as evolving neutrally[1]. It is now well-established that selection can act on synonymous sites, particularly in fast-growing species[2] and species with large effective population sizes ($N_e$), where efficient selection can optimise codon use to maximise translational velocity and accuracy[3,4]. However, it is not clear how pervasive constraint on synonymous sites is in the genomes of slower-growing and low-$N_e$ species, such as humans and other mammals, where evidence for codon usage bias due to translational selection is lacking or absent[5,6].

Several estimates of constraint at synonymous sites in mammals have been made; evidence for purifying selection acting on synonymous sites was found in 28% of mouse-rat orthologous genes[7], and a

genome-wide analysis estimated that ~25% and ~11% of fourfold degenerate (4d) sites are under constraint in hominids and murids respectively[8]. Evidence for selection on codons due to translational optimisation[9], protein folding[10,11], accurate splicing[12,13], and mRNA stability[14] is present in mammals[15]. In addition, overlapping regulatory features such as transcription factor (TF) binding sites[16], binding sites of RNA-binding proteins[17], or within upstream open-reading frames (uORFs)[18], may bring secondary functions to 4d sites. Selection for CpG dinucleotides, the predominant site of DNA methylation in mammals[19], in certain TFs and developmental genes also appears to lead to constraint at synonymous sites[20].

Vertebrate genomes are generally GC-rich at third codon positions ('GC3 content' hereon), which is in contrast to a genomic background of low GC content due to mutational bias, and may reflect a GC-rich vertebrate ancestral genome[21,22]. Neutral processes are likely to have had a strong influence on the GC content at synonymous sites, such as GC-biased gene conversion (gBGC) where allelic conversions in

[1]Department of Medical Biochemistry and Microbiology, Uppsala University, Uppsala, Sweden. [2]SciLifeLab, Uppsala University, Uppsala, Sweden. [3]Broad Institute of MIT and Harvard, Cambridge, MA, USA. ✉e-mail: matthew.christmas@imbim.uu.se

heteroduplex complexes formed during meiotic recombination are biased towards GC alleles[23–25]. Indeed, gBGC can result in signatures that 'mimic' those due to selection[26]. For example, the abundant and conserved mammalian stop codon TGA (TAA in most other taxa) is selectively sub-optimal and likely a result of gBGC[27], gBGC has likely led to the fixation of deleterious amino acid changes in primates[28], and translocation of the mouse *Fxy* gene into a highly recombining region led to rapid increase in GC3 content due to gBGC[29]. Recently it has been shown that gene methylation affects local mutation rates[30], and mutation rate heterogeneity can also result in different rates of sequence divergence throughout species' genomes independently from selection. The pervasive impacts of these neutral processes can therefore lead to high sequence conservation among species and need to be taken into account when investigating signatures of selection on codon usage[31].

Neutral and selective models of synonymous site evolution are not mutually exclusive and codon usage likely reflects a balance between selective and mutational pressures as well as drift[5,31–33]. Recently, the 'unwanted transcript hypothesis' (UTH) has been presented as a novel explanation for synonymous site GC bias and constraint in humans and other species with low $N_e$[32]. It proposes that selection on synonymous sites aids the retention of native, functional transcripts and the removal of spurious transcripts. Around 62% of the human genome is transcribed, yet only 1.22% encodes protein-coding exons[34]. Much of the transcription is likely spurious, generated from the integration of novel sequences into the genome such as from viruses and transposable elements (TEs), leading to costly transcripts whose removal is advantageous[32]. In species where $N_e$ is low, unwanted transcripts may accumulate due to inefficient selection against their mildly deleterious effects[35].

The UTH posits that much of the selection on synonymous sites relates to quality control of transcripts[32]. Synonymous sites can strengthen the signal that a transcript is native and differentiate it from spurious, non-functional transcripts. In the GC-poor genomes of mammals, native transcripts are generally GC-rich[21], CpG poor (due to hypermutability of methylated CpGs), contain introns, and have short exons (80% are <200 bp[36]). According to the UTH, transcripts that are AT-rich, have high CpG content, are intronless, and contain long exons are therefore most likely to be spurious and targeted for removal. In agreement with this, it has been shown that higher GC content at synonymous sites of transgenes in mammals leads to increased protein levels related to increased nuclear export[37–39], and intronless retrogenes tend to evolve higher GC content at 4d sites (GC4) than their intron-containing parent genes[37]. Selection on synonymous sites can therefore act to both reduce the production and increase the removal of spurious transcripts by ensuring accurate splicing and aiding their discrimination, respectively. Importantly, the UTH allows for a model whereby synonymous site GC bias has been driven by gBGC; if GC content is a signal of transcript nativity in mammals, then gBGC may preserve high GC content in native transcripts compared to spurious transcripts elsewhere in the genome[32], with gBGC acting predominantly in constrained, gene-rich regions[26,40] and opposing the GC → AT mutational bias.

In this study, we utilise an alignment of 240 placental mammal species genomes, produced as part of the Zoonomia project[41], to comprehensively characterise synonymous site conservation in mammals. We identify shared synonymous sites from the alignment and use single-base resolution phyloP scores generated in Zoonomia[42] to provide the most comprehensive characterisation and assessment of mammalian synonymous site conservation. We interpret our findings in the context of gBGC, mutation rate variation, and the UTH, advancing our understanding of the neutral and selective processes acting on synonymous sites in placental mammals.

## Results

### Conserved 4d sites are GC-biased

We identified all codons containing 4d sites in the human genome using Gencode annotation v.39[43] and taking the canonical transcript per gene ($n = 19,386$), revealing a total of 5,278,470 4d sites. Many of these are unlikely to be 4d sites in other mammal genomes due to differences in gene content. We therefore used the Zoonomia alignment[41] to identify codons where >85% of species were aligned and >95% of the aligned codons contained a 4d site. This resulted in a set of 2,621,118 4d sites distributed over 17,394 transcripts that formed the basis of our analysis (Supplementary Data 1). We extracted the base content of each 4d site within each aligned genome and used the single-base human-referenced phyloP scores generated in Zoonomia as estimates of conservation[42]. These scores range from −20 to 8.9, with negative scores indicating accelerated evolution, scores close to 0 indicating neutral evolution and positive scores indicating constrained evolution[44]. A significance threshold for constraint was previously established for this dataset using a 5% false discovery rate (FDR), where sites with a phyloP score ≥2.27 are under significant constraint[42]. While this 5% FDR threshold was previously used to identify functional sites likely under purifying selection (i.e. 'constraint'), here when considering synonymous sites we prefer to use the term 'conservation' so as not to imply selection. Overall, 20.8% of 4d sites have phyloP scores above this threshold (Table 1). In comparison, 29.4% of threefold (3d), 36.6% of twofold degenerate (2d) and 74.1% of nondegenerate sites show significant conservation at this threshold. Synonymous sites with a phyloP score above the 5% FDR threshold have a mean of 230 species aligned (s.d. = 9.16) and a mean of 188.5 species sharing the same base (s.d. = 37.8). We observe 4906 4d sites (0.19% of all 4d sites) that are fixed for the same base across all 240 genomes (24.3% fixed for A, 21.7% fixed for T, 27.3% fixed for C, 26.7% fixed for G).

All but one of the mammal genomes considered have a GC4 content >50% (Fig. 1a), with only the hispid cotton rat genome (*Sigmodon hispidus*, Order: Rodentia) below 50% (49.5%). The common shrew (*Sorex araneus*, Order: Eulipotyphla) has the highest GC4 content at 64.3%, likely due to their large $N_e$ and fast-evolving genomes[45]. When considering all 4d sites regardless of phyloP, GC content is only slightly higher than AT content in humans and across mammals (Table 1). However, over 80% of bases at 4d sites with phyloP ≥2.27 are G or C (Table 1), and we observe a positive correlation between GC4 content and conservation across the mammalian genomes (Fig. 1b–d; Pearson's $r = 0.28$, $P < 2.2 \times 10^{-16}$). Base content at conserved 4d sites is generally skewed towards a specific base in all mammals (Fig. 1e, f). We also see that transcript GC4 content correlates strongly between the human genome and mammal genomes generally (Fig. S1A; Pearson's $r = 0.90$, $P < 2.2 \times 10^{-16}$). To test whether 4d site conservation within transcripts relates to the GC4 content of transcripts, we compared the mean phyloP at 4d sites that are GC to the mean phyloP at 4d sites that are AT for each transcript. If 4d site conservation is largely driven by gBGC then we would not expect the two to correlate. However, we observe a strong positive correlation (Pearson's $r = 0.63$, $P < 2.2 \times 10^{-16}$; Fig. 1g), suggesting that whilst the majority of conserved 4d sites are GC, conservation within transcripts is not determined solely by gBGC.

### Codon-specific GC biases

The major allele in humans (identified from the genotypes of 138,922 individuals in the TOPMed dataset[46]) matches the most common base across the mammal genomes at 79.5% of all 4d sites, rising to 90.6% at conserved 4d sites. We generated base counts from the human genome across each of the codons containing 4d, 3d and 2d sites to assess whether GC bias is seen for all codons. Conserved 4d sites show a strong GC bias across all codons, with a minimum GC4 content of 59.5% (proline codons) and a maximum of 90.5% (valine codons; Fig. 2a). Differences in base proportions at 4d sites in the codons of each amino acid reflect context-specific restrictions on base content. For example, in codons where the second base is C (codons for alanine, proline, serine and threonine), 4d sites are rarely guanine, which would generate CpG sites. Indeed, 67% of conserved 4d sites in humans are G, and only 5% are C when the subsequent base is G. In glycine codons

**Table 1 | Summary of conservation at fourfold degenerate sites in mammals**

| Class | Number of 4d sites | Genomes | %A | %T | %C | %G |
|---|---|---|---|---|---|---|
| All 4d sites | 2,621,118 | Mammals[a] | 21.3 | 22.8 | 29.4 | 26.5 |
| | | Humans | 22.4 | 23.7 | 28.3 | 25.5 |
| **Conserved 4d sites** | | | | | | |
| Total | 544,503 (20.8% of all 4d sites) | Mammals | 9.8 | 9.6 | 39.9 | 40.8 |
| | | Humans | 9.6 | 9.4 | 39.8 | 41.1 |
| CpG site (in human genome) | 14,195 (2.61% of conserved sites) | Humans | – | – | 44.6 | 55.4 |
| **Conserved 4d sites near splice sites** | | | | | | |
| 5′ exon edge site (3′ splice site, position +1) | 484 (0.09% of conserved sites) | Mammals | 12.1 | 3.3 | 5.6 | 79.1 |
| | | Humans | 9.5 | 5.0 | 22.7 | 62.8 |
| 5′ exon edge site + 1 (3′ splice site, position +2) | 9571 (1.76%) | Mammals | 85.5 | 3.0 | 4.8 | 6.7 |
| | | Humans | 80.5 | 8.3 | 5.3 | 6.0 |
| 5′ exon edge site + 2 (3′ splice site, position +3) | 4167 (0.77%) | Mammals | 11.4 | 4.0 | 76.8 | 7.8 |
| | | Humans | 10.9 | 4.0 | 71.6 | 13.5 |
| 3′ exon edge site (5′ splice site, position −1) | 7341 (1.35%) | Mammals | 3.2 | 2.7 | 0.5 | 93.6 |
| | | Humans | 3.1 | 2.8 | 14.8 | 79.3 |
| 3′ exon edge site −1 (5′ splice site, position −2) | 3883 (0.71%) | Mammals | 14.1 | 34.0 | 22.1 | 29.8 |
| | | Humans | 14.7 | 33.3 | 23.8 | 28.2 |
| Alternative splice sites[b] | 1267 (0.23%) | Mammals | 9.9 | 9.7 | 33.5 | 46.9 |
| | | Humans | 9.5 | 9.8 | 36.1 | 44.6 |

[a]Percentages for mammals are based on the total counts of each base at each 4d site where >85% of the 240 species were aligned and the site was 4d in >95% of the codons.
[b]Splice sites not present in the canonical transcripts.

(GGN), 61% of the 4d sites are C, likely due to selection against GGG codons that can lead to the formation of G-quadruplex structures in mRNA[47]. However, if the subsequent base is G then 4d C content is low as this would generate CpG sites (38% A, 16% C, 32% G, 13% T). We also see strong biases for GC at conserved 3d and 2d sites in all codons except those for arginine. Conserved synonymous sites in all other codons are at least 72% C or G, with 74% C at conserved 3d sites in the codons for isoleucine (Fig. 2b, c). For sites with low conservation, we do not observe a general bias for higher GC content at synonymous sites (Fig. 2d–f).

### Higher GC4 content in single-exon genes and first exons

Mammalian transcripts typically contain introns. Under the UTH, native transcripts that are intronless should display high GC content at synonymous sites to aid their recognition as native transcripts despite lacking introns[32]. We tested this by comparing the proportion of 4d sites that are GC across the mammal genomes in single- ($n = 1119$) and multi-exon ($n = 15,483$) transcripts with >10 4d sites. Median GC4 in single-exon transcripts (0.67) is significantly greater than that in multi-exon transcripts (0.57; Wilcoxon rank-sum test, $W = 6,264,319$, $P < 2.2 \times 10^{-16}$; Fig. 3a). As first and last exons are rarely spliced out[48] it may be important that they maintain higher GC content as they are present in most transcripts and, as such, are most exposed to transcriptional silencing and degradation mechanisms, as well as containing essential splice sites. For exons with >10 4d sites, GC4 in the first exons of multi-exon transcripts are significantly higher than in other exons, (Fig. 3a; median GC4 in first exons = 0.72 ($n = 6070$), other exons = 0.58 ($n = 79,694$); Wilcoxon rank-sum test, $W = 157,517,032$, $P < 2.2 \times 10^{-16}$). Last exons also show significantly higher GC4 content, but the difference is smaller (median GC4 in last exons = 0.62 ($n = 8053$), other exons = 0.58 ($n = 78,830$); Wilcoxon rank-sum test, $W = 287,613,287$, $P < 2.2 \times 10^{-16}$). We observe a significant negative correlation between exon length and GC4 content for single-exon genes (Pearson's $r = -0.18$, d.f. = 1117, $P = 1.83 \times 10^{-9}$). This is predicted under the UTH as GC content is evidently most important towards the 5′ end for cytoplasmic localisation and translation of native transcripts[37].

### High conservation at splice sites

The UTH predicts high constraint at 4d sites involved in transcript splicing to avoid incorrect splicing that can lead to spurious transcripts. Our data revealed significantly higher phyloP at 4d sites positioned adjacent to splice sites compared to 4d sites elsewhere (Fig. 3b; $n = 1044$ (5′ exon boundary), 12,049 (3′ exon boundary), 2,608,026 (elsewhere in exons); mean phyloP = 1.82 (5′ boundary), 3.70 (3′ boundary), −0.17 (elsewhere); ANOVA, $F = 6561$, d.f. = 2, $P < 2.2 \times 10^{-16}$). A post hoc Tukey's test showed that phyloP at exon 3′ boundaries is significantly greater than at 5′ boundaries (difference in mean phyloP = 1.88, $P < 2.2 \times 10^{-16}$). For conserved 4d sites, we see strong biases in base content at or near splice junctions, likely reflecting selection for effective splicing. For example, in mammals there is a strong bias for G at both 5′ and 3′ exon boundaries, with 79.1% and 93.6% G observed, respectively (Fig. 3c–f). This has been shown previously at first and last exonic positions in strong splice sites[49,50]. Expanding the analysis to encompass 4d sites from human alternative splice junctions that are not used in the canonical transcripts, we see generally lower phyloP than 4d sites at constitutive splice junctions (mean phyloP = 0.03, s.d. = 3.79), but 1267 of them show significant conservation and also show a strong GC bias (Table 1).

### Human population variation at 4d sites

We used the TOPMed dataset of human genetic variation[46] to assess levels of variation at 4d sites in humans. We see that a lower proportion of conserved sites are variable compared to nonconserved sites (20.9% versus 28.3%) and minor allele frequencies (MAF) are significantly lower at conserved sites (mean MAF = $2.5 \times 10^{-4}$ and $6.9 \times 10^{-4}$ at conserved and nonconserved sites respectively; two-sample $t$ test, $t = 31.45$, d.f. = 1,465,717, $P < 2.2 \times 10^{-16}$). We calculated the proportion of 4d sites that were variable (MAF > 0) in phyloP bins and observed a significant negative correlation with phyloP (Fig. 4a). The relationship was more pronounced when considering the proportion of sites with more frequent variants (MAF > 0.001; Fig. 4b). These patterns are suggestive of stronger negative selection acting on conserved compared to other 4d sites. We also tested whether the proportions of variant types (i.e., major

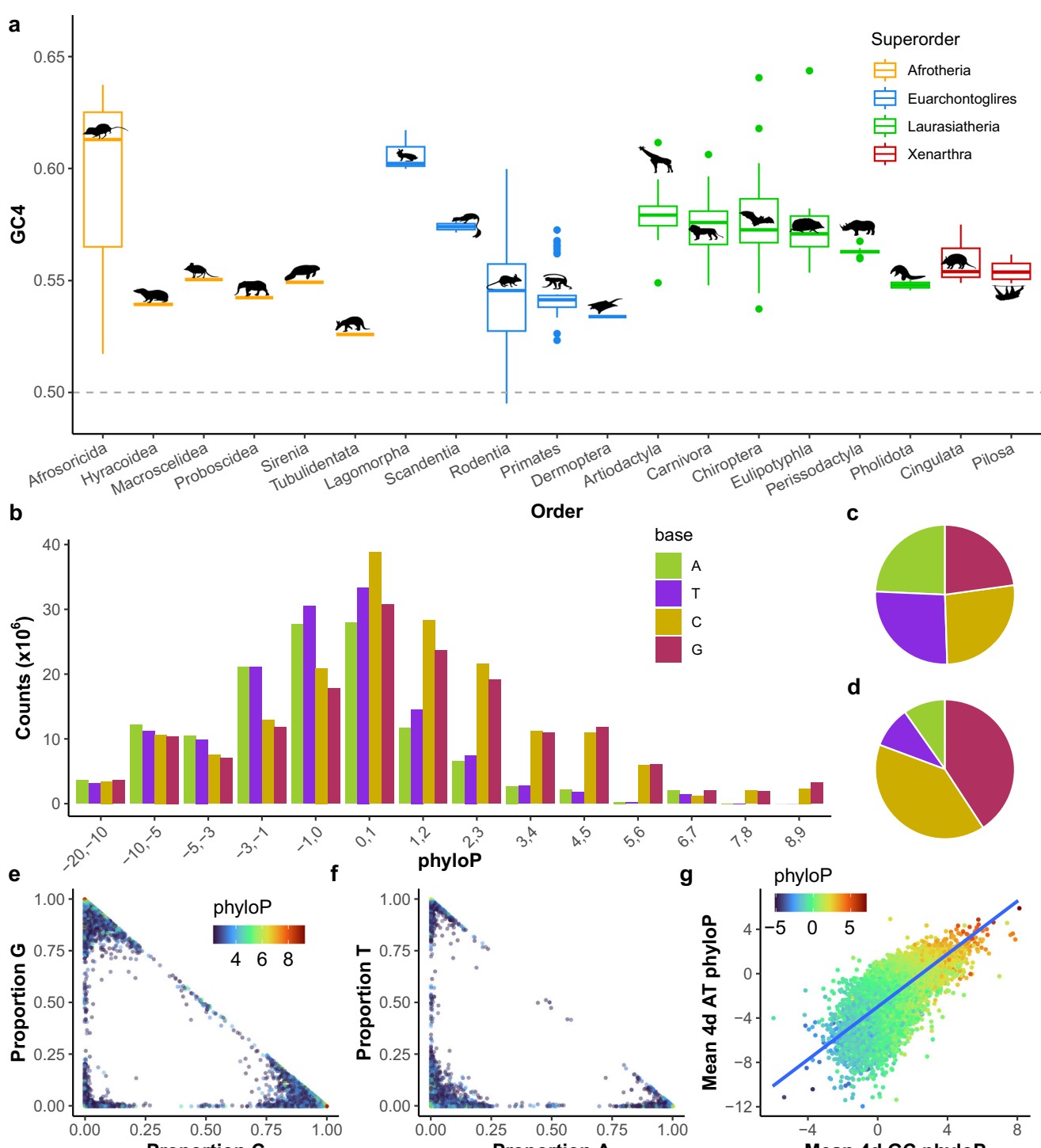

**Fig. 1 | Conservation at fourfold degenerate sites in mammals is GC-biased.**
**a** Boxplots showing genome GC content at 4d sites (GC4) per mammalian Order
($n = 240$ placental mammal genome assemblies). All mammal genomes in the
Zoonomia dataset show a GC4 bias except for the hispid cotton rat (*Sigmodon
hispidus*, Order: Rodentia; GC4 = 49.5%). Boxes represent first and third quartiles
with median line, whiskers extend ±1.5× IQR, outliers beyond whiskers are shown as
points. **b** The number of each base at 4d sites across 240 placental mammal gen-
omes, binned by phyloP score, where more positive scores indicate stronger con-
servation. Pie charts showing the proportion of each base seen at 4d sites across the

mammal genomes for sites **c** phyloP <2.27 and **d** phyloP ≥2.27. At conserved 4d sites
(phyloP ≥2.27), we observe a general bias towards C or G at GC sites (**e**) and a bias
towards A or T at AT sites (**f**) among the mammal genomes. **g** Per-transcript mean
phyloP at 4d sites that are GC positively correlates with mean phyloP at 4d sites that
are AT ($n = 17,394$; two-tailed Pearson's $r = 0.63$, $P < 2.2 \times 10^{-16}$). Blue trend line
shows a linear regression using the 'geom_smooth' function in ggplot2 in R. Colour
scale represents mean transcript 4d phyloP. Species silhouettes in (**a**) were
obtained from phylopic.org, which are available under a creative commons licence.
Source data are provided as a Source Data file.

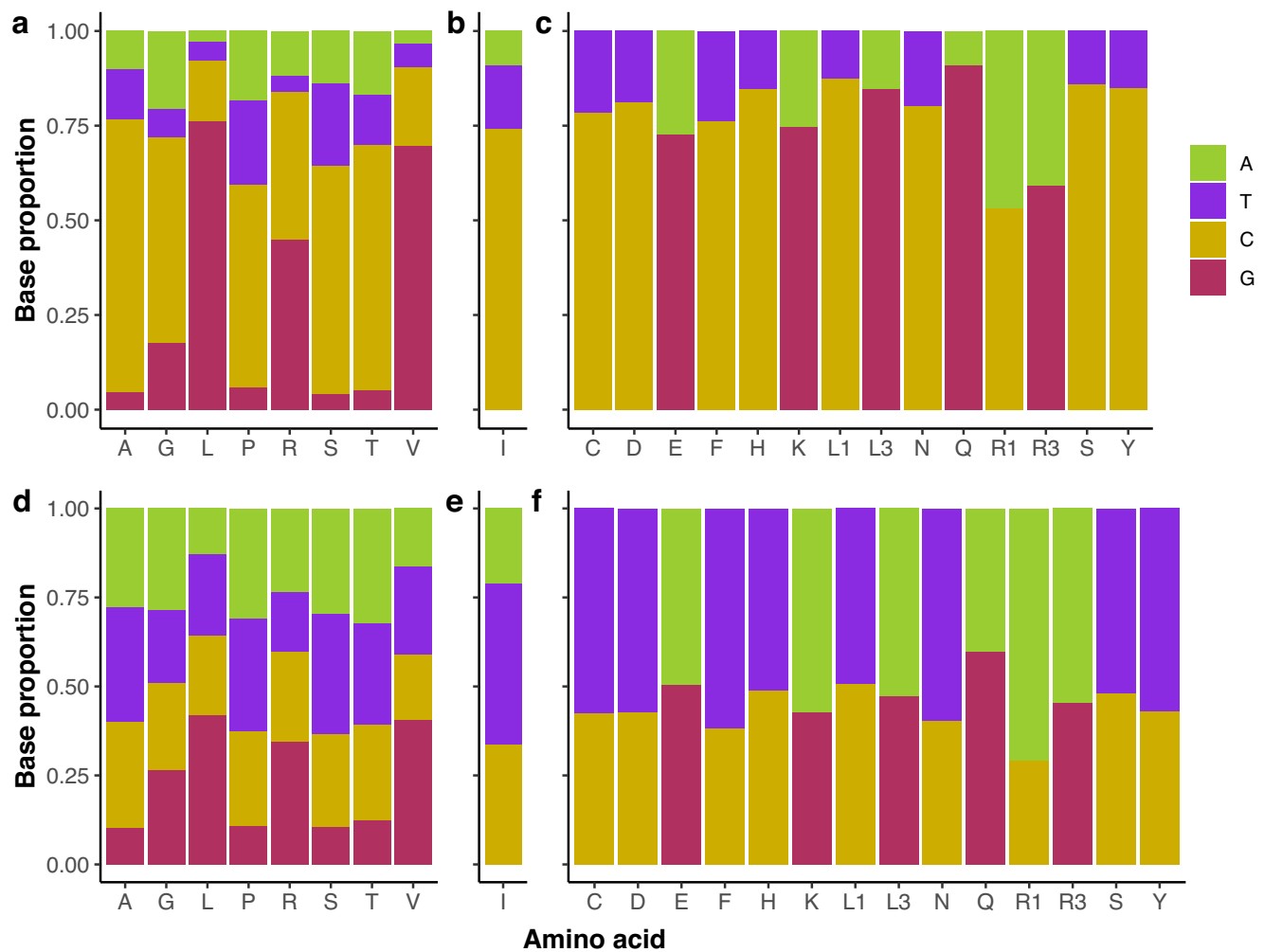

**Fig. 2 | Differences in G and C content at conserved fourfold degenerate sites likely reflect restrictions on codon usage.** Base content of significantly conserved **a** 4d ($n = 544,503$), **b** 3d ($n = 141,931$), and **c** 2d synonymous sites ($n = 2,287,435$) shows a strong GC bias for all amino acid codons in humans. Not significantly conserved **d** 4d ($n = 2,076,615$), **e** 3d ($n = 340,674$), and **f** 2d synonymous sites ($n = 3,946,958$) do not show such a bias. For conserved 4d sites, strong biases towards C or G bases largely reflect local base context, where sites that form CpGs are rare, as well as GGG codons that can lead to G-quadruplex structures in mRNA. Source data are provided as a Source Data file.

and minor allele combinations) at conserved sites were as expected if the variant type proportions are independent of phyloP. We found a significant difference between observed and expected variant type proportions (Pearson's $\chi^2$ test; $\chi^2 = 261$, d.f. = 11, $P < 2.2 \times 10^{-16}$). Specifically, conserved 4d sites show a deficiency in variable A and T sites and an excess of variable C and G sites (Fig. 4c), with A → G sites having the most negative and C → T sites the most positive residuals (−6.56 and 4.50, respectively; Fig. 4d). An excess of GC → AT polymorphisms is known in humans and can be explained by gBGC[24]. In the context of our results, this suggests that some of the high conservation we observe at CG 4d sites in mammals may have resulted from gBGC rather than purifying selection, whereas a deficiency of variation at conserved AT sites is more suggestive of selection.

**Exploring genome correlates with 4d site conservation and GC content**

We tested for correlates of GC4 content and phyloP in mammals to further explore evidence for selection versus neutral, mechanistic processes acting on 4d sites, such as gBGC and transcription-coupled repair[51]. We observed a significant positive correlation between transcript GC4 content across mammals and local GC content (measured in 1 Mb windows across the human genome; Fig. S1B; Pearson's $r = 0.52$, $P < 2.2 \times 10^{-16}$). The slope of the fitted linear model is 2.17,

demonstrating that genes with high GC4 content show considerably higher GC content than that of their neighbouring sequence. This is likely explained by the fact that the GC content of non-coding regions will be reduced by TE insertions[52], as well as a greater likelihood of recombination in conserved coding regions compared to non-coding regions[40], leading to a greater effect of gBGC in coding compared to neighbouring non-coding sequences. We found a negative correlation between local GC content and transcript 4d site mean phyloP (Fig. S1C; Pearson's $r = -0.22$, $P < 2.2 \times 10^{-16}$), showing that conservation of 4d sites is not associated with being in GC-rich genomic regions in the human genome. The GC4 content of transcripts also negatively correlates with mean 4d phyloP (Pearson's $r = -0.21$, $P < 2.2 \times 10^{-16}$; Fig. S1D). In addition, for transcripts with a mean 4d phyloP above the significance threshold of 2.27, there is no relationship between GC4 content and mean phyloP (Pearson's $r = -0.005$, $P = 0.91$; Fig. S1D), further suggesting that high conservation at 4d sites is not purely driven by gBGC. Transcript mean 4d site phyloP is positively correlated with phyloP at 0d sites (Pearson's $r = 0.42$, $P < 2.2 \times 10^{-16}$), with a distinct elevation in mean 4d phyloP in the transcripts with the highest 0d phyloP (Fig. S1E). We also observed significant positive correlations between transcript mean 4d site phyloP and UTR mean phyloP (Pearson's $r = 0.47$ and 0.53 for 5′ and 3′ UTRs respectively, $P < 2.2 \times 10^{-16}$ in both cases; Figs. S1F, G).

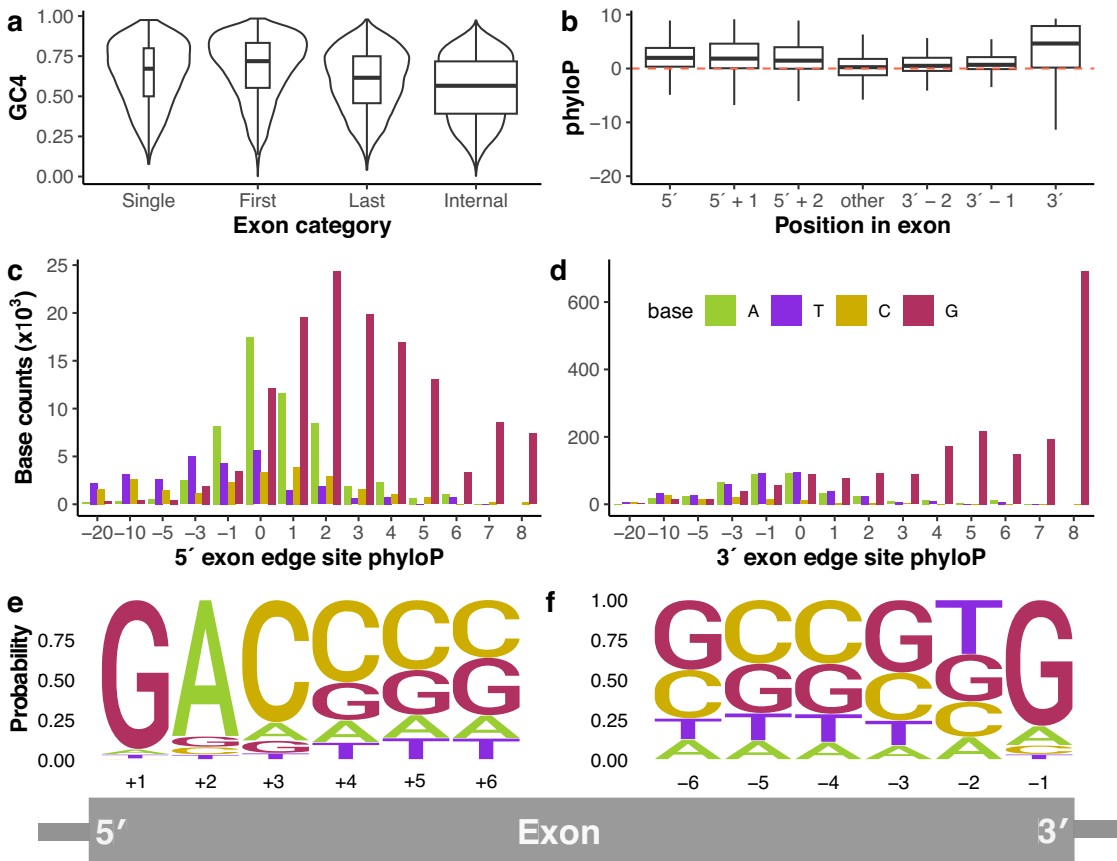

**Fig. 3 | GC content of 4d sites in mammalian transcripts supports the unwanted transcript hypothesis. a** GC4 content of single-exon genes ($n = 1119$) and the first ($n = 6070$) and last ($n = 8053$) exons of multi-exon genes are significantly greater than GC4 content of internal exons (two-sided Wilcoxon rank-sum tests, $P < 2.2 \times 10^{-16}$ in all cases). Violin plots show data density. **b** PhyloP is significantly higher at 4d sites within 3 bp of exon-intron boundaries ($n = 81,221$) compared to 4d sites elsewhere in exons ($n = 2,539,897$; ANOVA, $F = 6561$, d.f. = 2, $P < 2.2 \times 10^{-16}$). Red dotted line indicates neutrality. Boxplots in (**a**, **b**) show first and third quartiles with median line, whiskers extend ±1.5× IQR. Histograms of base counts across mammals at 4d sites at (**c**), the first (5′ exon edge, $n = 1044$) and **d** last (3′ exon edge, $n = 12,049$) positions in exons in phyloP bins show strong bias for G at conserved 4d sites. Logo plots for base content of 4d sites at (**e**), the first six positions ($n = 29,871$) and **f** last six positions ($n = 19,792$) in exons, where the probability reflects the counts of each base across the mammal genomes at 4d sites located at those positions. Grey diagram of exon puts the positions into the visual context of the exon/intron boundaries. Source data are provided as a Source Data file.

If high GC4 content at 4d sites is driven by transcription-coupled repair then it would be expected that GC4 content should relate to expression level, with highly expressed genes being subject to greater transcription-coupled repair[51]. To test this, we compared human gene expression data from multiple tissues and developmental stages[53] to 4d site phyloP and GC4 content. We show that, in general, GC4 content negatively relates to expression levels, with lowly expressed genes showing higher GC4 content on average than highly expressed genes (Fig. S2; mean GC4 content of lowly and highly expressed genes = 0.59, and 0.54, respectively, ANOVA, d.f. = 2, $F = 440.9$, $P < 2.2 \times 10^{-16}$). In addition, lowly expressed genes enriched for highly conserved 4d sites have significantly higher GC4 content than those that are highly expressed (Fig. S2; mean GC4 content of lowly and highly expressed genes = 0.70, and 0.59 respectively, ANOVA, d.f. = 2, $F = 829.3$, $P < 2.2 \times 10^{-16}$). These findings suggest that high expression and related transcription-coupled repair cannot explain the high GC4 content we observe at conserved 4d sites.

**Associations between synonymous site GC content, effective population size and genome transposable element content**

The UTH predicts that greater GC content of synonymous sites is more important for species with low $N_e$, where inefficient selection can lead to a greater load of spurious transcripts[32]. We used mammalian historical $N_e$ estimates generated as part of Zoonomia[54] ($n = 210$ species) to test for a relationship between GC4 content and $N_e$. Mean $N_e$ across the

210 species is 37,000 (s.d. = 39,000), with a ~1600-fold difference between the species with the highest $N_e$ of 270,000 (prairie deer mouse, *Peromyscus maniculatus*, Order: Rodentia), and the lowest $N_e$ of 162 (beluga whale, *Delphinapterus leucas*, Order: Cetartiodactyla). We do not see a significant correlation between $N_e$ and GC4 generally (Pearson's $r = -0.13$, $P = 0.073$; Fig. S3A). However, we do observe positive correlations in three Orders, namely Artiodactyla, Carnivora, and Primates (Fig. S3B). This fits the gBGC model, where a higher equilibrium GC content would be expected in larger populations[55].

To assess whether $N_e$ is related to GC content of highly conserved 4d sites, we reduced the data to a set of 4d sites that are represented in all 240 genomes (i.e., no missing data; $n = 93,464$ sites, mean phyloP = 3.84, s.d. = 1.32). We then estimated genetic distance among all species at these sites and generated a neighbour-joining tree (Fig. 5a). The tree recapitulates the expected topography of the placental mammal tree[56], showing that divergence at these 4d sites is generally lineage-specific. GC content at these sites negatively correlates with historical $N_e$ (Fig. 5b; Pearson's $r = -0.48$, $P = 5.3 \times 10^{-13}$) as well as mean genetic distance to all other species (Fig. 5c; Pearson's $r = -0.87$, $P < 2.2 \times 10^{-16}$). The Indochinese shrew (*Crocidura indochinensis*, Order: Eulipotyphla) is the most divergent species, having the lowest GC content as well as the third highest $N_e$ in the dataset (~169,000). Contrastingly, the edible dormouse (*Glis glis*), a rodent with an exceptionally low $N_e$ of 520 shows the highest GC content at these constrained 4d sites among rodents (74%), and cetaceans have

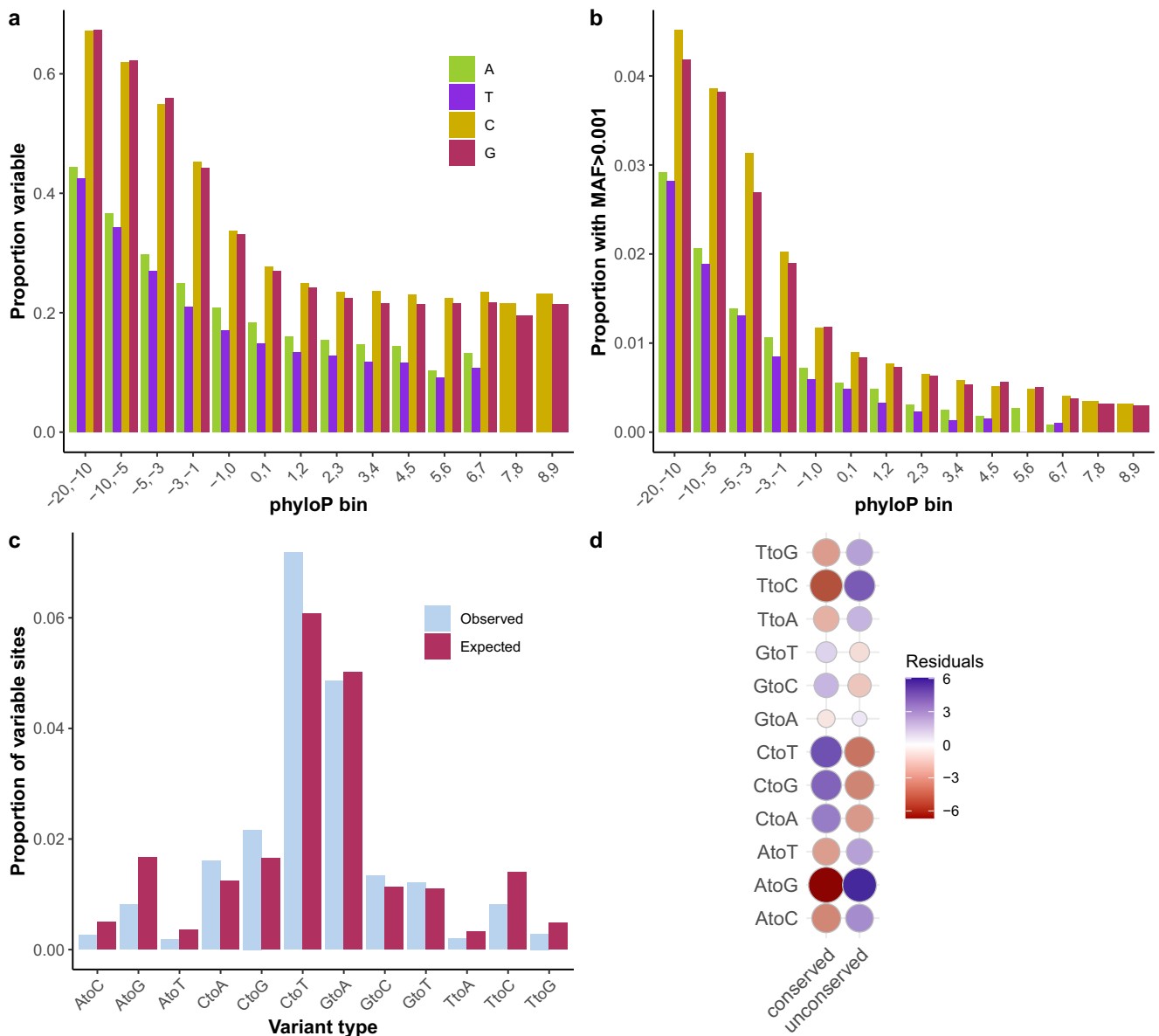

**Fig. 4 | Human variation at fourfold degenerate sites is negatively associated with phyloP score. a** The proportion of 4d sites that are variable (minor allele frequency >0; $n = 716,661$) based on single-nucleotide polymorphism data from TOPMed data freeze 8 was calculated in phyloP bins. For all four bases, the proportion of variable sites is significantly negatively correlated with mean phyloP in each bin (two-tailed Pearson's $r = -0.86, -0.91, -0.74$, and $-0.80$, $P = 0.012, 0.0048, 0.022$, and $0.010$ for A, T, C and G bases, respectively). **b** Same as (**a**) but considering only variants with a minor allele frequency >0.001 ($n = 27,818$). Again, all four bases show a significant negative correlation with mean phyloP in each bin

(two-tailed Pearson's $r = -0.94, -0.92, -0.98$, and $-0.97$, $P = 0.0019$, $0.0038, 4.9 \times 10^{-6}$, and $1.04 \times 10^{-4}$ for A, T, C and G bases, respectively). **c** When considering only significantly conserved 4d sites (phyloP > = 2.27; $n = 544,503$), we observed a significant difference in the proportions of variant types compared to expected proportions based on total numbers of each variant type independent of phyloP (two-tailed Pearson's Chi$^2$ test; $\chi^2 = 261$, d.f. = 11, $P < 2.2 \times 10^{-16}$). **d** Residuals from the Chi$^2$ test show that there is a deficiency in variable A and T sites and an excess of variable C and G sites amongst conserved 4d sites. MAF minor allele frequency. Source data are provided as a Source Data file.

amongst the highest GC content at these sites as well as the lowest historical effective population sizes. We also see a negative correlation between genome size and GC4 content at these 4d sites (Pearson's $r = -0.25$, $P = 8.9 \times 10^{-5}$; Fig. S4A). This pattern is particularly pronounced when looking within certain groups, including Primates, Eulipotyphla, and Muridae rodents (Fig. S4B–D), and fits with a model of gBGC, where chromosome length is inversely related to recombination rate and, therefore, the effect of gBGC[45].

Genome transposable element content may indicate (1) the efficiency of selection to purge unwanted/non-native sequences and (2) the burden of unwanted transcripts on a genome. A lower effective population size may result in higher genome TE content[35] (although see ref. 57), which may subsequently lead to stronger selection on

synonymous site GC content, as the problem of unwanted transcripts is elevated[32]. Despite this prediction of the UTH, we do not see a significant negative correlation between total genome TE content and $N_e$ (Pearson's $r = -0.10$, $P = 0.14$). However, when looking only at young TEs (based on low divergence from consensus sequences[58]), which may be more indicative of the burden of active TE-induced unwanted transcripts in a genome, the correlation is significant (Fig. 5d; Pearson's $r = -0.20$, $P = 0.0034$). There is also a significant positive correlation between genome young TE content and GC4 content when considering all 4d sites (Fig. 5e; Pearson's $r = 0.16$, $P = 0.016$). We also looked for associations within orders, as levels of young TE content vary considerably among and within mammalian orders[58]. Both Rodentia (Fig. 5f; $n = 53$, Pearson's $r = 0.29$, $P = 0.035$) and Chiroptera (Fig. 5g;

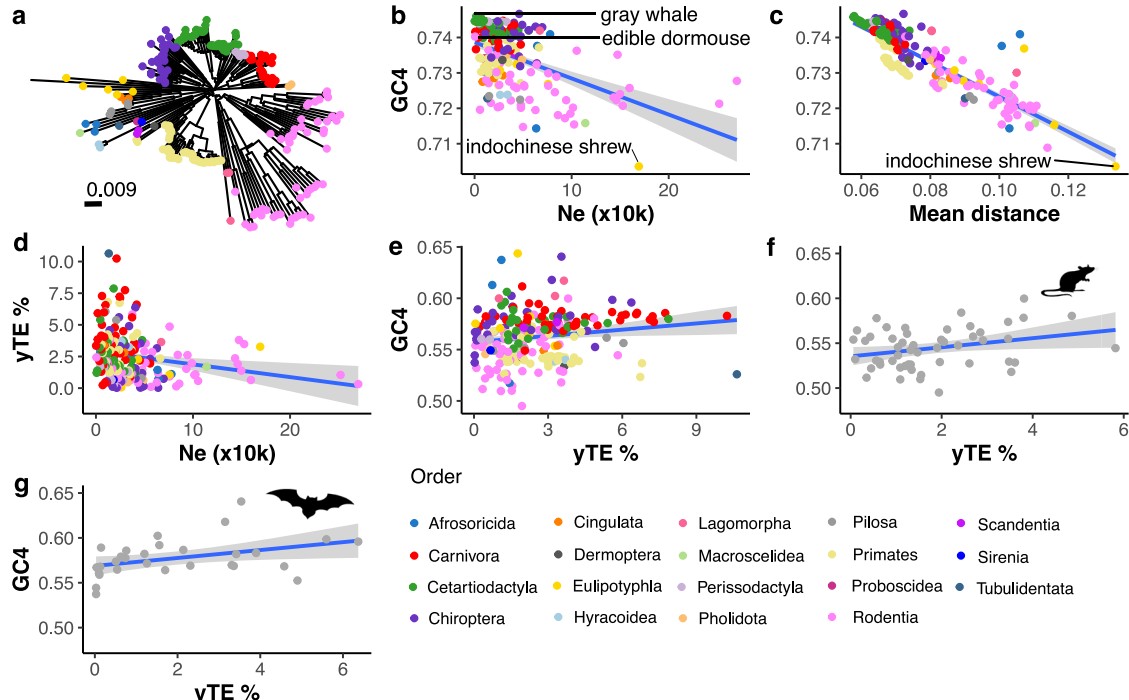

**Fig. 5 | Effective population size and genome young transposable element content relates to GC4 content. a** Neighbour-joining tree of 240 mammals based on genetic distances (sequence dissimilarity) calculated from 93,464 shared 4d sites. Significant negative correlations are seen between GC4 content at highly conserved sites and **b** historical $N_e$ (two-tailed Pearson's $r = -0.48$, $P = 5.3 \times 10^{-13}$) and **c** mean genetic distance of a species to all other species (two-tailed Pearson's $r = -0.87$, $P < 2.2 \times 10^{-16}$). Indicated in (**b**), the edible dormouse (*Glis glis*, Order: Rodentia) has the lowest $N_e$ and highest GC4 content of rodents, the grey whale (*Eschrichtius robustus*, Order: Artiodactyla) has the highest GC4 content of all species, and the Indochinese shrew (*Crocidura indochinensis*, Order: Eulipotyphla) has the greatest mean distance to all other species, the lowest GC content at conserved 4d sites, and the third highest $N_e$. Genome young TE (yTE) content (a proxy for unwanted transcript burden) negatively correlates with historical $N_e$ (**d** two-tailed Pearson's $r = -0.20$, $P = 0.0034$) and positively correlates with GC content at 4d sites (**e** two-tailed Pearson's $r = 0.16$, $P = 0.016$). The two orders with the strongest correlations between yTE content and GC4 content are Rodentia (**f** $n = 53$, two-tailed Pearson's $r = 0.29$, $P = 0.035$) and Chiroptera (**g** $n = 30$, two-tailed Pearson's $r = 0.40$, $P = 0.031$). Trend lines in (**b–g**) were calculated using a linear model with standard error intervals around the mean shown in grey. Species silhouettes in (**f**) and (**g**) were obtained from phylopic.org, which are available under a creative commons licence. Source data are provided as a Source Data file.

$n = 30$, Pearson's $r = 0.40$, $P = 0.031$) show significant positive correlations between young TE genome content and GC4 content, as predicted under the UTH.

## Distribution of conserved 4d sites among genes

Across the 17,394 transcripts within which we identified high-confidence 4d sites shared among mammals, 16,860 (96.9%) contain at least one and 12,257 transcripts (70.5%) have 10 or more conserved 4d sites. Excluding splice-related sites within 3 bp of exon boundaries, there are still 11,832 transcripts (68.0%) with 10 or more conserved 4d sites. The top 1% most highly conserved 4d sites (phyloP ≥7.13, $n = 26,282$, GC4 = 99%) are distributed over 7893 transcripts (45.4%), with 4475 transcripts (25.7%) having at least two and 503 transcripts (2.9%) at least 10 highly conserved 4d sites. The gene containing the topmost 1% highly conserved 4d sites is *TNRC6B* (Trinucleotide Repeat Containing Adaptor 6B) with 113 (13.1% of its 865 4d sites). Within this gene, 60% of its 4d sites are significantly conserved ($n = 519$, mean phyloP = 5.63, GC4 = 54.9%), with 72 sites that are fixed for the same base among all aligned genomes (GC4 = 61%). We assessed enrichment for conservation by comparing observed and expected rates of conserved 4d sites (Fig. 6a). There are 7197 transcripts (43.4%) with observed/expected rates greater than 1, demonstrating that 4d site conservation is not distributed evenly among transcripts. The top 95th percentile of conservation enrichment (observed/expected rate >1.98) contains 853 transcripts, and 166 transcripts are in the top 99th percentile (observed/expected rate >2.83; Supplementary Data 2). Gene ontology analysis revealed strong enrichment for functions relating to development and transcriptional regulation (Supplementary Data 3).

The gene with the highest enrichment for 4d site conservation was *IGIP*, a single-exon gene coding for immunoglobulin A inducing protein (19/19 4d sites conserved, GC4 = 58%). Sixteen out of the 39 mammalian Homeobox (HOX) genes (41%) are within this top 99th percentile of 4d site conservation enrichment, and 24/39 (62%) are within the top 95th percentile.

CpGs are generally depleted in mammalian genomes due to the hypermutability of methylated CpGs, except for CpG islands in the promoter regions of some genes[59]. We observe that 10% of all 4d sites form CpGs in the human genome. Conservation at 4d sites in human CpGs is significantly less than at other 4d sites ($n = 265,111$ and $2,357,007$, mean phyloP = −3.98 and 0.28 at 4d sites in and not in CpGs respectively; two-sample $t$ test, $t = 441.23$, d.f. = 293,603, $P < 2.2 \times 10^{-16}$). This likely reflects the hypermutability of CpG sites and transcripts with high CpG content are known to be suppressed by zinc-finger antiviral protein (ZAP)[60]. However, 14,195 4d sites in CpGs show significant conservation (2.6% of conserved sites), which likely relates to selection acting on epigenetic regulation and repression of spurious intra-gene body transcription of certain genes via methylation[20,61].

We calculated the enrichment of conserved 4d CpG sites within transcripts and observed a subset of genes enriched for high 4d conservation in CpGs (Fig. 6b). The top 99th percentile (observed/expected ratio >9.00) contains 115 genes (Supplementary Data 2), 41 of which are also in the top 99th percentile of genes enriched for 4d conservation generally. Gene ontology enrichment analysis of the top 99th percentile of CpG-enriched genes revealed enrichment for similar functions to the general 4d site conservation set, including developmental processes and transcriptional regulation (Supplementary

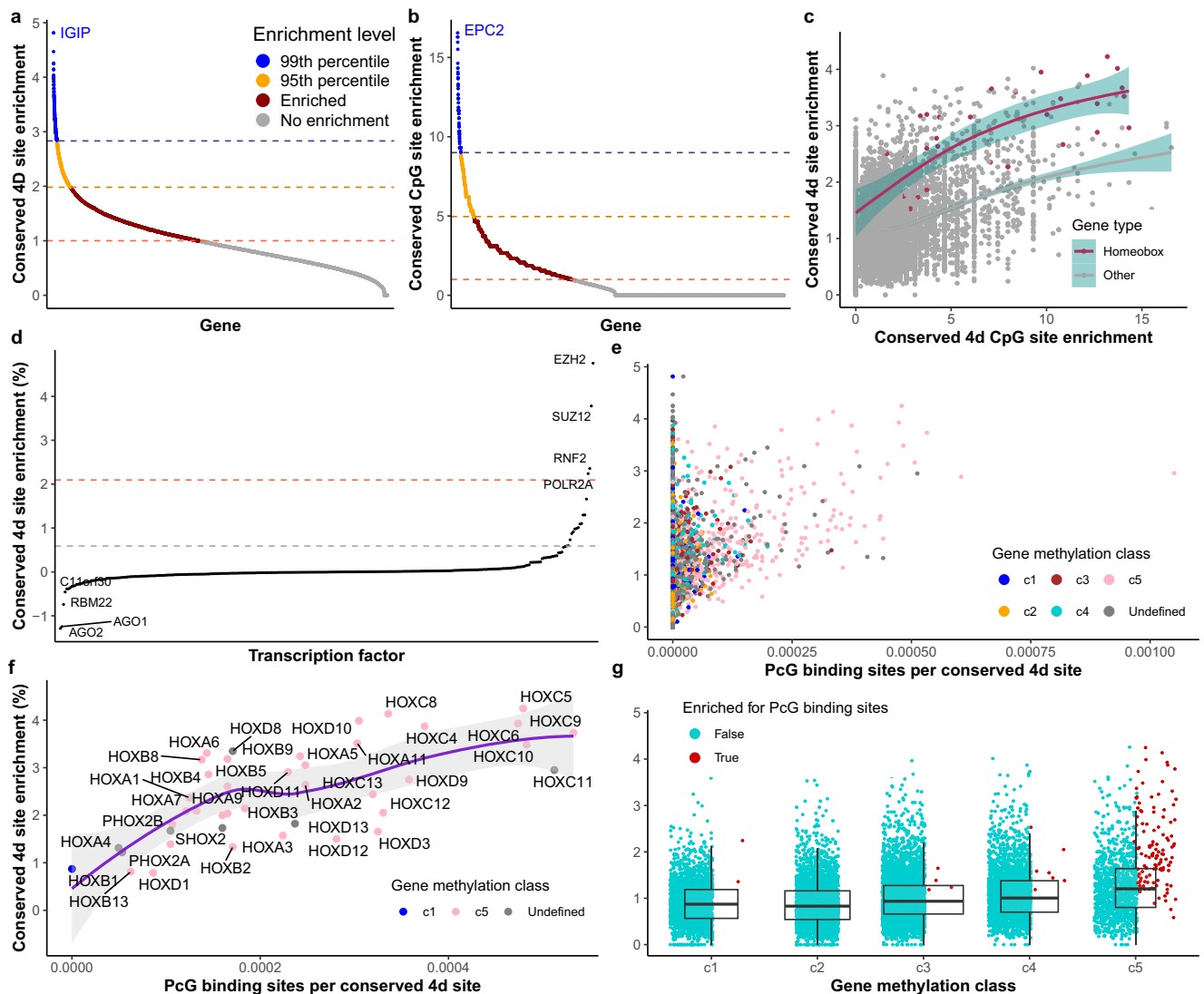

**Fig. 6 | Gene enrichment of conserved fourfold degenerate sites relates to Polycomb group protein binding and hypomethylation. a** We calculated enrichment for conserved 4d sites by comparing the observed to the expected rate of conserved sites within transcripts ($n = 16,602$ transcripts). Transcripts with ratios >1 show evidence for enrichment. The top 95th and 99th percentiles of enrichment are indicated (orange and blue). **b** Enrichment was also calculated for conserved 4d sites within CpGs for each transcript with at least five 4d sites within CpGs ($n = 11,428$ transcripts). **c** Enrichment of conserved 4d sites (excluding CpG sites) and conserved 4d CpG sites within transcripts positively correlates (two-tailed Pearson's $r = 0.42$, $P < 2.2 \times 10^{-16}$). The correlation is even stronger when considering only the homeobox group of developmental transcription factors ($n = 41$ transcripts; two-tailed Pearson's $r = 0.78$, $P < 1.9 \times 10^{-9}$). Smoothed trend lines were generated using the 'geom_smooth' function in ggplot2 with the 'gam' method with 95% confidence interval around mean values in cyan. **d** Enrichment of conserved 4d sites in the binding sites of human transcription factors, measured using ENCODE 3 transcription factor ChIP-seq peaks ($n = 326$ transcription factors). Binding sites of Polycomb group (PcG) transcription factors *EZH2, SUZ12*, and RNF2, as well as RNA Polymerase II Subunit A (*POLR2A*), show strong enrichment for conserved 4d sites.

Dashed lines represent the top 95th (grey) and 99th (red) percentiles of enrichment. **e** Enrichment of conserved 4d sites positively correlates with overlap of PcG protein-binding sites in transcripts ($n = 16,617$ transcripts; two-tailed Pearson's $r = 0.24$, $P < 2.2 \times 10^{-16}$). Genes are coloured by DNA methylation classes (from Mas-Ponte and Supek, 2024), where genes in the same class share similar local DNA hypomethylation profiles. The correlation in (**e**) is mostly driven by genes in methylation class c5. Genes in this class show hypomethylation along the whole length of the gene. **f** The correlation is strongest amongst Homeobox genes, nearly all of which are in methylation cluster c5 ($n = 41$ transcripts; two-tailed Pearson's $r = 0.68$, $P = 7.3 \times 10^{-7}$). The smoothed trend line was generated using the 'geom_smooth' function in ggplot2 with the 'loess' method with 95% confidence interval around mean values in grey. **g** Enrichment of conserved 4d sites within genes relates to hypomethylation, with genes in c5 having significantly greater enrichment of conserved 4d sites than genes in other methylation classes ($n = 15,170$ transcripts; ANOVA, d.f $= 4$, $F = 162.4$, $P < 2.2 \times 10^{-16}$). This gene class also contains 89.7% of genes enriched for PcG protein-binding sites (red). Boxes represent first and third quartiles with median line, whiskers extend ±1.5× IQR. Source data are provided as a Source Data file.

Data 4). The gene *EPC2* (enhancer of polycomb homologue 2) had the highest enrichment with 8 out of 9 4d sites in CpGs conserved (enrichment score = 16.6). EPC2, a highly conserved Polycomb group (PcG) protein, is predicted to play an important role in the regulation of chromatin structure in eukaryotes[62]. The PcG family of proteins are involved in many cellular memory processes, including cell fate decisions during embryogenesis and maintaining the transcriptional repressive state of genes over successive cell generations via

H3K27me3 histone modifications[63–65]. The gene *CNOT2* (CCR4-NOT transcription complex subunit 2) also shows high enrichment for conserved CpG sites (enrichment score = 16.3). The CCR4-NOT complex interacts with *TNRC6B*, the gene with the most strongly conserved 4d sites, to regulate mRNA synthesis and degradation[66]. HOX genes also show high prevalence of conserved 4d CpG sites, with 14 in the top 99th percentile of enriched genes. For example, *HOXC9* has 54 4d sites within CpGs, 40 of which are conserved.

## Conservation at 4d sites relates to PcG binding and methylation

There is a positive correlation between enrichment for conserved CpGs at 4d sites and enrichment for conserved 4d sites (after removing CpG sites) in transcripts (Pearson's $r = 0.42$, $P < 2.2 \times 10^{-16}$; Fig. 6c). This pattern is more pronounced when considering only HOX genes (Pearson's $r = 0.78$, $P = 1.9 \times 10^{-9}$; Fig. 6c). This is expected under the UTH if selection on synonymous sites modulates RNA structure to physically 'hide' CpG dinucleotides in native transcripts from anti-CpG quality control measures such as ZAP[32,60]. In addition, it has recently been shown that within-gene mutation rate heterogeneity can be explained by methylation levels, with hypomethylated regions also being hypomutated[30]. In particular, genes enriched with the Polycomb silencing histone mark H3K27me3, many of which are developmental genes including HOX genes, are characterised by hypomethylation and hypomutation throughout their gene bodies[30]. This made us consider the possibility that mutation rate variation within and among genes may explain some of the variation in 4d site conservation, potentially explaining the positive correlations we see between enrichment for conserved 4d sites and conserved 4d CpG sites, as well as between phyloP at 4d GC sites and 4d AT sites within genes (Fig. 1g).

To assess a potential association between 4d site conservation and mutation rate variation, we first looked to see whether genes targeted by PcG transcriptional repressors (hallmarks of hypomethylated genes[30,67,68]) are enriched for conserved 4d sites. We measured overlap between 4d sites and ENCODE 3 TF ChIP-seq clusters[69] representing 340 TFs from 129 cell types and calculated an enrichment score based on the number of overlaps between binding sites and conserved 4d sites for each TF. We found that the majority of TFs show no enrichment for binding to conserved 4d sites, with 60% of the TFs having an enrichment score below 0.01 and 95% an enrichment score below 0.59 (where 0 = no enrichment; Fig. 6D). However, four genes showed strong enrichment for conserved 4d sites in their TFBS ChIP signals, (99th percentile, enrichment score >2.1), three of which are PcG genes (*EZH2*, enrichment score = 4.75; *SUZ12*, enrichment score = 3.78; and *RNF2*, enrichment score = 2.36), as well as *POLR2A* (enrichment score = 2.24), revealing an association between PcG protein binding and 4d conservation. To test this further and identify genes where the association between PcG protein binding and 4d conservation is strongest, we then calculated the number of PcG-binding sites per conserved 4d sites for all genes and found a significant positive correlation with gene enrichment of conserved 4d sites (Pearson's $r = 0.24$, $P < 2.2 \times 10^{-16}$; Fig. 6e). Despite this correlation, there is a subset of genes strongly enriched for conserved 4d sites that show no or very little binding by PcG proteins, demonstrating that high 4d site conservation is not only found in genes targeted by PcG proteins. Genes in the top 99th percentile of conserved 4d site-PcG protein-binding site overlap ($n = 168$ genes; Supplementary Data 5) are enriched for GO terms relating to transcriptional regulation and embryonic morphogenesis and development (Supplementary Data 6). Genes absent of binding sites for the three polycomb group proteins but enriched for conserved 4d sites ($n = 109$, Supplementary Data 7) are enriched for GO terms relating to the regulation of gene expression and metabolic processes (Supplementary Data 8).

We next considered a more comprehensive classification of gene methylation by using the five gene methylation classes (c1–c5) recently developed by Mas-Ponte & Supek[30], where all genes within a class share similar methylation profiles. Briefly, genes in c1 have methylated promoters, low expression and a depletion in active transcription chromatin states, c2 genes are similar to c1 but show some promoter hypomethylation as well as active chromatin states, c3 and c4 are active, expressed genes with wider hypomethylation including the promoter and extending ~2 kb of the TSS. Genes in c5 have overall less methylated gene bodies across their whole length, overlap repressed TSS and enhancer states, and are enriched for the Polycomb silencing histone mark H3K27me3[30]. Using these methylation classes, we saw

that the majority of genes driving the positive correlation between gene enrichment of conserved 4d sites and the number of PcG-binding sites per conserved 4d site are hypomethylated genes in class c5 (Fig. 6e). For HOX genes, which are predominantly in the c5 methylation class, this positive correlation is even stronger (Pearson's $r = 0.68$, $P = 7.3 \times 10^{-7}$); Fig. 6f). Interestingly, *HOXB1*, which does not contain any PcG protein-binding sites and is the only HOX gene in methylation class c1, has one of the lowest conserved 4d site enrichment scores among the HOX genes (score = 0.87).

We compared the conserved 4d site enrichment scores between genes in each of the five classes and found significantly greater enrichment for conserved 4d sites in gene classes with more extensive gene hypomethylation (increasing from c1 to c5; ANOVA, d.f. = 4, $F = 162.4$, $P < 2.2 \times 10^{-16}$; Fig. 6g). Genes in the c5 class, which have the most extensive hypomethylation, are more enriched for conserved 4d sites than all other classes (Tukey's HSD, $P = 5.5 \times 10^{-10}$ in all four comparisons) and this class contains almost all (89.7%) of the genes enriched for PcG-binding sites.

## Enrichment of conserved 4d CpG sites relates to gene expression

The UTH predicts that highly expressed genes may show higher intra-gene body methylation to suppress the generation of unwanted transcripts. To address this, we looked at whether genes enriched for conserved 4d CpGs—the main sites of methylation—show greater expression than other genes. Mean gene expression levels from Cardoso-Moreira et al.[53], of genes enriched for CpG conservation are significantly higher than in other genes, both within and across the seven organs in the dataset (Fig. S5; mean RPKM expression across all organs = 11.6 and 24.3, s.d. = 258 and 81.5 for genes lacking CpG enrichment and genes in the 99th percentile of CpG conservation enrichment respectively; ANOVA, d.f. = 3, $F = 24.44$, $P = 8.28 \times 10^{-16}$; Tukey's HSD $P = 2.45 \times 10^{-12}$). Conserved CpG-enriched genes are also transcribed in significantly more cells in the mouse organogenesis cell atlas[70] (Fig. S6; mean cell counts = 69,915 and 178,020 for genes lacking CpG enrichment and conserved CpG-enriched genes respectively; ANOVA, d.f. = 2, $F = 195.9$, $P = <2.2 \times 10^{-16}$; Tukey's HSD $P = 7.65 \times 10^{-9}$). This supports the predictions of the UTH, where intra-gene body methylation in highly expressed genes in regions of open chromatin can act to suppress the generation of unwanted transcripts[61] and suggests selection acting on these CpG sites.

## Conservation at 4d sites relate to overlapping regulatory features

We performed a general linear model between 4d site phyloP and 11 potential constraining features, as well as other potential correlates (background GC content, synonymous site mutation rate; Table 2) for all 4d sites with a phyloP ≥0 ($n = 1,663,871$) to test how much of the observed conservation can be explained by these features. Sites with negative phyloP scores were excluded to test only for predictors of sequence conservation and not acceleration. The model explained 9.7% of the variation in phyloP and all but one variable (overlap with microRNA (miRNA) binding sites, $P = 0.20$) significantly explained some of the variance at $P < 0.05$. An insignificant effect of miRNA binding is not surprising given that it has previously been shown that only a very low proportion of synonymous site conservation is attributable to miRNA pairing[71]. The variable that explained most of the variance, with a relative importance of 85.5%, was GC proportion across mammals. Distance to the exon boundary (2.7% relative importance) and the background GC content (2.1% relative importance) were the next most important variables in the model. The remaining ten variables had a combined relative importance of 10.1%, suggesting that a significant but limited amount of the conservation observed as synonymous sites across mammals can be explained by being within conserved regulatory features such as transcription factor

**Table 2 | Outputs from a general linear model of synonymous site conservation in mammals**

| Variable | Estimate | Std. error | t value | Pr(>|t|) | Relative importance |
|---|---|---|---|---|---|
| Intercept | −0.54 | 0.078 | −6.84 | $7.68 \times 10^{-12}$ | – |
| GC proportion | 1.46 | 0.0041 | 353.51 | $<2.2 \times 10^{-16}$ | 0.86 |
| Distance to exon edge | −0.00037 | $6.0 \times 10^{-6}$ | −61.93 | $<2.2 \times 10^{-16}$ | 0.027 |
| Background GC proportion | −1.71 | 0.035 | −49.08 | $<2.2 \times 10^{-16}$ | 0.021 |
| Within cCRE | 0.15 | 0.0041 | 36.98 | $<2.2 \times 10^{-16}$ | 0.020 |
| Number of exons | −0.0054 | 0.00015 | −35.35 | $<2.2 \times 10^{-16}$ | 0.015 |
| Within TFBS | 0.13 | 0.0048 | 27.88 | $<2.2 \times 10^{-16}$ | 0.014 |
| Within uORF | 0.013 | 0.00032 | 41.48 | $<2.2 \times 10^{-16}$ | 0.014 |
| CpG site | −0.50 | 0.0088 | −57.19 | $<2.2 \times 10^{-16}$ | 0.014 |
| Synonymous site μ | 4392.50 | 164.16 | 26.76 | $<2.2 \times 10^{-16}$ | 0.0069 |
| Number of aligned species | 0.010 | 0.00031 | 32.04 | $<2.2 \times 10^{-16}$ | 0.0053 |
| Within ESE 70 bp from exon end | 0.20 | 0.0058 | 35.49 | $<2.2 \times 10^{-16}$ | 0.0047 |
| Within RBP-binding site | 0.0036 | 0.00013 | 28.25 | $<2.2 \times 10^{-16}$ | 0.0029 |
| Within lncRNA | 0.018 | 0.0053 | 3.44 | $5.75 \times 10^{-4}$ | 0.00031 |
| Within miRNA-binding sites | −0.0039 | 0.0030 | −1.29 | 0.20 | – |

Model $r^2 = 0.097$.

binding sites, candidate *cis*-regulatory elements (cCREs), RNA-binding protein (RBP) binding sites, or exonic splicing enhancers (ESEs).

Among conserved sites, we used a similar general linear model to test whether feature overlap related to gene expression, potentially revealing differences in the selection pressures acting on synonymous sites in lowly and highly expressed genes. Whilst the model was significant (Supplementary Data 9), it had an $r^2$ of only 0.038, showing that very little of the variation in feature overlap relates to gene expression (Fig. S7A). Synonymous sites in lowly expressed genes have a much lower overlap with RBP-binding sites (relative importance of 97.5% in the model) compared to those in more highly expressed genes (Fig. S7B). However, very little of the conservation at 4d sites is explained by overlap with RBP-binding sites (Table 2) and this result likely reflects that binding of RBPs, which are involved in transcript processing[17], relates to expression levels independent of constraint. The model therefore did not reveal any evidence that selection pressures acting on 4d sites differ between highly and lowly expressed genes.

## Discussion

We have carried out a comprehensive characterisation of conservation at synonymous sites in mammalian genomes using the Zoonomia 240-species genome alignment and single-base resolution phyloP scores[42], providing the most complete picture to date of synonymous site conservation over the 100 million years of placental mammal evolution. All species except the hispid cotton rat (*Sigmodon hispidus*) show a GC bias at 4d sites across their genomes. We show that conservation at 4d sites strongly relates to GC content, with the vast majority (~80%) of highly conserved 4d sites in mammals being G or C. We demonstrate that the GC4 content of human transcripts correlates strongly with that across mammals generally, suggesting that similar processes have shaped synonymous site base content across mammals. Whilst high sequence similarity across species is often used to imply purifying selection and evolutionary constraint[42,72], neutral processes such as gBGC and mutation rate variation can also lead to high sequence conservation. In this discussion, we will consider the evidence we find for selection acting on 4d sites against neutral processes, and place our findings in light of the recently presented UTH.

Codon usage bias is a ubiquitous phenomenon in nature and mammals share similar codon usage frequencies with a bias towards GC at codon third positions[73,74]. One hypothesis proposed to explain this is that the ancestral vertebrate genome was GC-rich, which would

have created a prevalence of GC-ending codons early in vertebrate evolution[21]. Substantial increases in synonymous site GC content in certain mammalian lineages over the last 100 million years have been related to differences in the strength of gBGC among species[45]. As the impact or extent of gBGC in a genome is the product of the effective population size by the recombination rate by the repair bias[26], larger effective population sizes and smaller chromosomes are predictors of a higher equilibrium GC content and a faster increase in GC content under the gBGC model[45,55]. Our findings fit with these expectations, where we see variable GC4 content among the mammal genomes that relates positively with historical $N_e$ in some orders, and negatively with genome size (as a proxy for chromosome size) overall.

Our findings also support the gBGC model in that we observe a positive correlation between GC content of 4d sites and neighbouring non-coding sequences. Many genes have GC4 contents that are considerably higher than their neighbouring sequences and 4d site conservation does not relate to local GC content, which may be more suggestive of selection than gBGC. However, non-coding neighbouring sequences are more vulnerable to insertions by AT-rich TEs, which over time can result in a differential between the GC content of coding and surrounding non-coding sequences[52]. In addition, we see an excess of mutations at conserved CG 4d sites in humans, which is suggestive of a lack of purifying selection on these sites and fits with the gBGC model[24].

Whilst our findings suggest that high ancestral GC content and the gBGC model can explain much of the GC4 content observed in mammals, in agreement with previous studies[24,26,31,52], there are several observations we make that cannot be so easily explained by gBGC. In our dataset, there are negative correlations between 4d site conservation and transcript GC4 as well as local GC content. This suggests that conservation is lower in regions more recently affected by gBGC, likely due to the fact that recombination landscapes vary greatly across the genomes of different species, with recombination hotspots evolving rapidly[75]. For example, hotspot conservation is low between humans and chimpanzees[76] and local recombination rates correlate only very weakly between strains of mice[77]. We show that 4d site conservation is strongest in genes whose proteins are under high constraint in mammals and therefore have experienced strong purifying selection throughout mammalian evolution. Repeated exposure to gBGC throughout the histories of synonymous sites in these ancient genes could then lead to elevated and conserved GC4 content[40]. However, whilst this may be true for some genes, we show that

transcript GC4 content is not a good predictor of 4d site conservation, where we observe a whole spectrum of GC4 proportions among the genes with high conservation at their 4d sites. In addition, we observe a positive correlation between 4d phyloP at AT and GC sites in transcripts, which would not be expected if conservation was driven by gBGC. An explanation of transcription-coupled repair leading to higher GC4 content is also not supported by our results, as genes with high GC4 content and conservation actually show lower levels of expression than other genes.

We find several lines of evidence that conservation at synonymous sites may relate to unwanted transcript mitigation, lending support to the UTH. The strong GC bias observed at conserved 4d sites may help 'flag' transcripts as native[32]. Higher GC content of 4d sites in single-exon genes and first exons of multi-exon genes, a pattern previously shown in humans[37] and shown here to be true across mammals, is also predicted by the UTH to differentiate native from unwanted transcripts. We also show strong evidence for high constraint at 4d sites located at splice sites, essential for ensuring the production of accurate transcripts[37,78–80]. Whilst 4d sites at exon boundaries are generally highly conserved, they explain <3% of the overall observed 4d conservation. We find evidence that secondary functions, such as 4d sites being located within TFBSs, uORFs, RBP-binding sites, and ESEs, can explain only a limited amount of the observed 4d site conservation.

A recent analysis of selection against point mutations in the human genome found no evidence for ultraselection (complete absence of point mutations) at 4d sites but that 39% experience weak purifying selection[79]. Our analysis of human genetic variation at 4d sites also provides evidence of purifying selection acting on some sites. Human variation at 4d sites correlates negatively with phyloP, and the 21% of 4d sites identified as under significant conservation show significantly less variation than other 4d sites, both in terms of the proportion of sites containing variants and the allele frequencies at variable sites. Whilst there is an excess of variants at conserved GC 4d sites, which is expected under the gBGC model[24], there is a deficiency of variants at conserved AT 4d sites, suggestive of purifying selection. Variants at synonymous sites have also been shown to be rare close to exon ends, likely due to disruption of ESEs[81].

The UTH predicts that spurious transcription should be a greater issue for species with low $N_e$ as selection is not efficient enough to remove weakly deleterious mutations that lead to spurious transcripts[32]. Whilst we did not see a relationship between GC4 content and $N_e$ across all species, and a positive correlation within some orders as predicted by the gBGC model, we did find that species with lower $N_e$ generally have higher GC content at 4d sites present across all 240 genomes. Under the UTH, this may be interpreted as a higher constraint on GC4 content at these sites in mammals with lower $N_e$ to mitigate spurious transcription[32]. In addition, it may reflect faster evolutionary rates in species with high $N_e$ and more efficient selection on codon use, although this is not widely observed in mammals[45,82]. We also saw that GC4 content positively correlates with the young TE content of a genome, which can be a source of spurious transcription, with elevated GC4 content in species with a potentially greater unwanted transcript load. This is despite the fact that young TE content correlates positively with genome size, which has a negative association with GC4 content. This is most strongly observed in bats and rodents, where variation in young TE content is greatest. Selection for error-mitigating properties in the error-prone genomes of species with small $N_e$ has been previously suggested as an explanation for the observation of an association between $N_e$ and intronic content and splice site usage across species[83]. Our findings support this hypothesis further, where higher GC4 content may act to limit the impact of spurious transcription and improve the detection of native transcripts amongst profuse unwanted transcripts[32].

Recently, it has been shown that mutation rate heterogeneity at the sub-gene scale in humans is caused by methylation variability, with hypomethylated regions having low mutation rates[30]. Reduced mutation rates in certain genes and gene regions will lead to elevated conservation over time, which can be reflected in synonymous site divergence. Such mutation rate heterogeneity may partly explain the positive correlation we see between phyloP at GC and AT 4d sites within transcripts, as well as the negative correlation between phyloP and genetic variability in humans. It is known that certain genes and gene regions show strong hypomethylation (so-called 'DNA methylation valleys'[84]), many of which are developmental genes associated with the Polycomb repressive mark H3K27me3[30,67]. Mas-Ponte and Supek have now demonstrated that these genes experience lower mutation rates due to their hypomethylation[30]. We have shown that many genes enriched for conserved 4d sites and CpG sites are also hypomethylated developmental genes targeted by PcG proteins. These genes encode amongst the most conserved mammalian proteins under strong evolutionary constraint[42], including HOX genes whose highly orchestrated expression regulates embryo development along the anterior-posterior axis[85]. We suggest that the high 4d site conservation in these genes likely relates to the lower mutation rates they have experienced throughout mammalian evolution compared to other genes.

DNA methylation is an important mechanism regulating the expression of developmental genes, such as HOX genes, located within Polycomb-associated DNA methylation valleys (reviewed in ref. 86). Sites of methylation in these genes, including CpG sites, are therefore likely to be under strong selection to ensure their accurate epigenetic regulation in time and space[20]. We show that many developmental genes are enriched for conserved CpG sites, despite the hypermutability of methylated CpGs leading to low CpG content of native transcripts generally. An enrichment of conserved CpG sites in the exons of HOX and other developmental genes leading to intragenic methylation[20] could serve to protect gene bodies from rogue entry of RNA polymerase II, which can lead to spurious expression[61,87]. Irregular methylation of these genes, often related to PcG gene misregulation and altered H3K27me3 deposition[88], has been associated with developmental disorders[68,89,90], cancer[86,88] and ageing[91–93]. A potential association between synonymous site constraint and ageing is intriguing; CpG methylation at or near HOX genes is a strong predictor of longevity in mammals[91] and changes within the mammalian chromatin landscape associate with aberrant transcription initiation inside genes during senescence and ageing[87]. Higher conservation of 4d sites in the developmental genes of larger, longer-lived mammals suggested in our data may reflect higher constraint on epigenetic regulatory features, including intragenic methylation. Whether this can facilitate longer life by reducing the effects of spurious transcription makes for an intriguing future line of research.

In conclusion, we show that synonymous site conservation in mammals is extensive and has likely arisen through a combination of several non-mutually exclusive processes, including gBGC, mutation rate variation, and selection. Highly constrained, integral genes involved in developmental processes and transcriptional regulation are enriched for conserved 4d sites, likely resulting from a combination of experiencing low mutation rates and strong selection on 4d sites relating to tight epigenetic regulation. We present some evidence in support of the prediction of the unwanted transcript hypothesis that species with small $N_e$ and high spurious transcript burden have greater GC4 content, which may help to differentiate native from unwanted, spurious transcripts.

## Methods

### Genome alignment and phyloP scores

We used the 241-way multi-genome Cactus[94] alignment of placental mammal genomes and human-referenced (Hg38) phyloP scores generated as part of the Zoonomia project[42]. PhyloP scores were generated across all autosomes using the PHAST package[44,95] with inputs of

the Cactus alignment and ancestral repeats as a model of neutrally evolving sequences. PhyloP scores were generated at single-base resolution and ranged in value from −20 to 8.9. Positions showing significant conservation were identified by converting phyloP values to $q$ values using a FDR correction. Any positions with positive phyloP scores and a $q \leq 0.05$ were considered to be significantly conserved. This gave a significance threshold of phyloP $\geq 2.27$ at 0.05 FDR. Further details can be found in the original paper[42]. PhyloP scores are available on the UCSC genome browser.

### Transcript selection and identification of mammalian 4d sites

Gene annotations for Hg38 were obtained from GENCODE release 39[43]. As most genes contain multiple transcripts, we chose an approach to select one representative transcript per protein-coding gene to simplify analyses. We selected representative transcripts using the Matched annotation from NCBI and EMBL-EBI set of representative transcripts (MANE Select)[96]. If unavailable here, then we selected the canonical transcript used by gnomAD[97] or BUSCO[98]. We used BEDtools v.2.30.0 intersect[99] to extract phyloP scores for the CDS of each transcript from the single-base phyloP score files. The degeneracy of each site within CDS was assessed by looking up each codon in a DNA codon table using a custom Perl script, revealing a total of 5,278,470 4d sites. Many of these are unlikely to be 4d sites in other mammal genomes due to, for example, differences in the gene content among genomes. We therefore extracted alignments for all codons from the Zoonomia alignment and identified codons where >85% of species were aligned and >95% of the aligned codons contained a 4d site (i.e. each 4d site is present in at least 194 species). This resulted in a set of 2,621,118 4d sites distributed over 17,394 transcripts that formed the basis of our analysis. Base content per 4d site per species were then extracted from the alignment and summarised using custom Perl scripts. Whether a fourfold degenerate site forms a CpG site was determined from the human reference genome using a custom Perl script which identified all 4d Cs followed by a G and all 4d Gs preceded by a C. Exon edge sites, exon counts per transcript, and single-exon genes were identified from the GENCODE annotation. GC content in 1 Mb windows was measured across the human reference genome (Hg38) using a custom Perl script with a bed file as output. We then used the 'intersect' tool of BEDTools v.2.30.0[99] between this file and a bed file of transcripts to obtain a local GC percentage for each transcript.

### Human population variation at 4d sites

We used the NHLBI Trans-omics for precision medicine (TOPMed) data freeze 8 (https://topmed.nhlbi.nih.gov/) containing whole genome sequencing of 138,922 individuals to assess human genetic variation at 4d sites. We used the 'intersect' tool of BEDTools v.2.30.0 to intersect the TOPMed VCF file and our 4d site bed file. At each 4d site with variants, we ascertained whether or not the allele in the human reference genome (Hg38) matched the major allele in the human population using a custom Perl script. This allowed us to identify variants that were specific to Hg38 and ensure that we were considering the most common allele when looking at base content of human 4d sites.

### Effective population sizes, transposable elements, and divergence at 4d sites

Estimates of historical $N_e$ for 210 of the Zoonomia species were obtained from Wilder et al.[54], where they used PSMC[100] to estimate $N_e$ based on heterozygous positions in each genome. Data on genome transposable element content was obtained from Osmanski et al.[58], where they annotated the TE content of all Zoonomia genomes including identifying young insertions as TEs with sequences with Kimura two-parameter genetic distances <4% compared to their consensus[58]. Genome size was taken as the total assembly size of each genome in the alignment. To analyse divergence at highly conserved 4d sites we first identified a set of sites where all 240 species were aligned, giving 93,464 4d sites with no

missing data. We then created concatenated pseudosequences in fasta format for each species from the alignment using a custom perl script. A single fasta format file containing these pseudosequences for all species was then read into R using the package Biostrings (https://doi.org/10.18129/B9.bioc.Biostrings). We then calculated a genetic distance matrix from the sequences using the R package DECIPHER (https://doi.org/10.18129/B9.bioc.DECIPHER), where each value in the matrix is the dissimilarity between two sequences. A neighbour-joining tree was then generated based on these distances using the Ape package in R. The tree was visualised using ggtree in R (https://doi.org/10.18129/B9.bioc.ggtree).

### Conservation enrichment

For each transcript, we calculated the observed versus expected proportion of conserved 4d sites, conserved 4d sites in CpGs, and TFBS overlap with conserved 4d sites. In each case, the expected proportion was calculated by 1/(total number of events/number of conserved events) (i.e. total 4d sites/total conserved 4d sites, total 4d CpG sites/total conserved 4d CpG sites, total TFBS-4d site overlaps/total TFBS-conserved 4d site overlaps). For each transcript, the observed rate was calculated by 1/(total events/conserved events). We then divided the observed by the expected proportion to get an enrichment score. We used the quantile function in R to identify highly enriched genes in each case, taking the top 95th and 99th quantiles. Gene ontology enrichment analysis was run on gene sets enriched for 4d conserved, 4d CpG conserved, and with conserved 4d sites enriched in TF binding sites using PANTHER[101] with a Bonferroni correction for multiple testing. To assess whether gene expression relates to conservation, we made use of three gene expression datasets: the GTEx gene expression data for 54 human tissues[102] (https://gtexportal.org/home/), the Evo-Devo mammalian organs dataset[53] (https://apps.kaessmannlab.org/evodevoapp/), and single-cell expression data from the mouse organogenesis cell atlas[70] (https://oncoscape.v3.sttrcancer.org/atlas.gs.washington.edu.mouse.rna/landing). We tested for associations between mean RPKM expression scores and levels of conservation- and conserved CpG enrichment using analysis of variance tests, as well as for associations between conserved CpG enrichment and the number of cells expressed in using the mouse organogenesis cell atlas.

### Functional annotation datasets

Various datasets were used to look for relationships between 4d site conservation and functional annotations. All human cCREs were obtained from ENCODE SCREEN. Per-gene synonymous site mutation rates were obtained from supplementary dataset 11 in the gnomAD paper[97]. miRNA targets were obtained from TargetScanHuman release 8.0 (https://www.targetscan.org/vert_80/) and converted from Hg19 to Hg38 coordinates using the UCSC liftover tool (https://genome.ucsc.edu/cgi-bin/hgLiftOver). uORF coordinates were retrieved from http://github.gersteinlab.org/uORFs/[18] and converted to Hg38 using liftover. RNA-binding protein-binding sites were obtained from ENCODE eCLIP signals detailed in ref. 103 and available from https://www.encodeproject.org/. Transcription factor binding sites were obtained from ENCODE 3 TF ChIP-seq datasets here: https://hgdownload.soe.ucsc.edu/goldenPath/hg38/encRegTfbsClustered/. Assignment of human genes to five distinct methylation classes was obtained from Mas-Ponte & Supek[30]. Hg38 coordinates of constrained TFBSs were provided by the authors of[42]. For identifying putative exonic splicing enhancers we downloaded the 238 hexamers identified as candidate ESEs in humans[104] from http://hollywood.mit.edu/burgelab/rescue-ese/ as well as the INT2 intersect set of 316 ESE motifs from Cáceres and Hurst (2013) and removed redundancy from the two sets. We then used a custom Perl script to check whether each 4d site was inside one of the hexamers to define whether a 4d site was inside or outside ESEs. We included only ESEs if they were within 70 bp of exon ends where they are most likely to be involved in splicing and therefore under

constraint[13]. Indeed, we do see higher 4d site conservation in ESE motifs within 70 bp of exon ends compared to elsewhere in an exon (mean (sd) phyloP = 0.26 (3.40) and −0.19 (3.58) for 4d sites in ESEs within 70 bp of exon ends and elsewhere respectively; Welch two-sample $t$ test, $t = -29.68$, d.f. = 130937, $P < 2.2 \times 10^{-16}$).

## Statistical analysis

All statistical analysis was carried out in R v.4.3.1 "Beagle Scouts". We used the 'cor.test' function to calculate Pearson's r for correlation analyses. We performed two-tailed $t$ tests (two groups) or ANOVA (>2 groups) to test for differences between means using the 't.test' and 'aov' functions, respectively. Post hoc Tukey's honestly significant difference tests were used to identify significant differences following ANOVA. A Chi$^2$ test was used to test for differences in observed versus expected proportions of variant types at conserved 4d sites using the 'chisq.test' function. We used the 'glm' function to perform a general linear model to test for associations between 4d site phyloP and multiple factors that may influence sequence conservation. For all conserved 4d sites, we used a GLM to test whether differences in gene expression (calculated as mean expression over all tissues and developmental stages from the Cardoso-Moreira et al.[53], dataset) relate to 4d site overlap with regulatory features.

## Reporting summary

Further information on research design is available in the Nature Portfolio Reporting Summary linked to this article.

## Data availability

The Cactus 241-way alignment and phyloP constraint scores are publicly available at https://cglgenomics.ucsc.edu/data/cactus/ and at https://genome.ucsc.edu/cgi-bin/hgTrackUi?db=hg38&g=cons241way. Mammalian 4d sites that formed the basis of the analysis are provided in supplementary data file 1 in Hg38-referenced bed file format. An R data object containing all data and associated scripts used to analyse the data is available from figshare via the following https://doi.org/10.6084/m9.figshare.26318410. Source data are provided with this paper.

## Code availability

Scripts detailing all of the analyses are available from figshare via the following https://doi.org/10.6084/m9.figshare.26318410.

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

## Acknowledgements

Computations and data handling were enabled by resources in projects NAISS 2023/5-90 and NAISS 2023/6-193 provided by the National Academic Infrastructure for Supercomputing in Sweden (NAISS), partially funded by the Swedish Research Council through grant agreement no. 2022-06725. K.L.T. is a Distinguished professor funded by the Swedish Research Council.

## Author contributions

K.L.T. and M.J.C. conceived the project; M.J.C. and M.X.D. carried out the bioinformatics; M.J.C. performed the analyses; M.X.D., J.R.S.M., S.V.K. and K.L.T. contributed to interpretation; M.J.C. wrote the first draft and all authors edited the manuscript.

## Funding

## Competing interests

The authors declare no competing interests.
