## [Transparent Peer Review file · Nature Communications]

Interpreting mammalian synonymous site conservation in light of the unwanted transcript hypothesis

Corresponding Author: Dr Matthew Christmas

Version 0:

Reviewer comments:

Reviewer #1

(Remarks to the Author)
Review of Christmas et al.

In this timely paper, the authors combine the recent zoonomia 240 species alignment with the even more recent unwanted transcript hypothesis (UTH). The latter is an attempt to resolve the enigma of commonplace selection on synonymous sites in mammals that have small effective population sizes and hence – according to the nearly neutral hypothesis – should have inefficient selection and hence an absence of selection on synonymous mutations. The UTH squares this circle by arguing that species with low N_e have in turn an acute problem with unwanted transcripts and so selection favours devices to reduce their rate of creation (HUSH, selection on splice control, 5' RNA pol II extension, intragene methylation to suppress intragenic transcription) or filter them out. Both, but especially the latter, can select for devices (export pathways, RNAses, P bodies etc) that favour GC rich transcripts (as mutation bias is GC->AT biased, neutral equilibrium is AT rich) this in turn favouring synonymous mutations that make a transcript appear more native.

The authors find abundant evidence that UTH makes a strong case for the constraint seen finding that constrained 4 fold sites tend to be GC rich, that intrapopulation variation suggests selection against GC reducing mutations, and that there is the predicted association with splice sites etc. The most striking result is that the trends run inverse to N_e as predicted by the model (an extension of Wu and Hurst's argument, as noted) and that recent TE insertions (the most likely to remain transcriptionally active?) predict the degree of constraint.

All in all the paper makes a strong case that the UTH is the framework within which to consider selection on synonymous sites in mammals. This then in broad terms is a valuable paper, not least because the common assumptions have been either a) that owing to low N_e there is no selection on synonymous sites in mammals (despite all the evidence to the contrary) or b) that codon usage bias is reflective of translational selection (seen in *E. coli*, yeast etc). I was also struck by the fact that gBGC appears not to explain the patterns.

In general, I have very few concerns about the quality of the analysis which appears to be very well done (and obviously to a significant scale). With all such analyses there are choices that could be different (I'll detail some of these). However, perhaps the biggest concern – although it isn't a big concern – is in the writing. I think it would be better to have some clear statements not just of results but to summarize the overall take. I was unsure in several places as to whether the authors considered the data in agreement with UTH or not. Indeed, the abstract could end with a definitive statement to the effect that "We conclude that the unwanted transcript hypothesis is the most viable model to explain the commonality of selection on synonymous sites in mammals" – or something to that effect.

Specific comments:

1. Specifically, in the section on CpG conservation, I wasn't clear what the authors think the relationship was with the UTH. The original UTH authors note that intragene body methylation is a strategy to reduce spurious intragene transcription (citing the current paper's ref 57, Neri et al). Neri et al indeed, note that intragene body methylation is there to suppress unwanted transcripts and is especially evident in the most highly expressed genes (those being in open chromatin much of the time we must presume and thus most vulnerable to spurious transcription). In this sense it makes perfect sense to me from the UTH point of view. However, there is also the problem that on the whole CpG is rare in human CDS so – according to UTH we

have systems like ZAP and RIG-I to filter CpG rich transcripts. In this sense, at first sight, the UTH seems to want to have it both ways: it can explain why high CpG is filtered out but also why some genes might select for conserved CpG. It isn't however totally plastic as it suggests follow-on predictions: when CpG is conserved these should be in highly expressed genes (a la Neri et al) and, as Radrizzani suggest when discussing UpA rich genes, the RNA structure and other synonymous sites should evolve to prevent the CpGs from being exposed to ZAP etc. This leads to two predictions: genes with conserved CpG should be highly expressed and the other synonymous sites should be under selection to force RNA structure in a manner than conceals the CpGs. The first is readily tested – the second not so easy. But you could ask whether if there is an abundance of CpG's conserved do we also see evidence for conservation at other synonymous sites in the same gene – as suggested by by Schattner, P. & Diekhans, M. Regions of extreme synonymous codon selection in mammalian genes. *Nucleic Acids Res.* 34, 1700–1710 (2006) in the context of UpA. This relates to my overall issue that the discussion could be stronger – it does seem to me that the UTH makes a remarkably strong case for what we see.

2. Line 86, Transcripts that are AT rich....I think you might preface this saying "According to the UTH, transcripts... "Otherwise it isn't clear whether this is your prediction or theirs.

3. Line 149 – reflecting the hypermutability... according to UTH it could also be because we have filters like ZAP against CpG sites.

4. Line 152 selection for epigenetic regulation – is regulation the only possibility? Surely as with Neri et al it could simply be suppression of intragene body spurious transcription?

5. Line 212 and on – I couldn't see that the hypothesis predicts a GC by exon length effect beyond avoidance of A3 in HUSH susceptible long exons (as evidenced in Radrizzani et al's figure 3). Indeed, the original HUSH papers find the effect is not so much preference for GC as avoidance of A. For single exon genes I would expect a negative correlation as the GC preferences seem to be very much on the 5' ends (Mordstein et al *Cell Systems*). Assuming the 5' end necessary for this effect is a fixed size, the GC4 should be lower for longer transcripts. Mechanistically this may relate to the PolII extension effect: Vlaming, H., Mimoso, C. A., Field, A. R., Martin, B. J. E. & Adelman, K. Screening thousands of transcribed coding and non-coding regions reveals sequence determinants of RNA polymerase II elongation potential. *Nat. Struct. Mol. Biol.* 29, 613–620 (2022) but we don't yet know enough about RNA export and the role of the 5' CDS end.

6. Line 234 – these splice junction proximal effects are likely to reflect selection on splice junctions more than GC content aren't they – how do they relate to splice site strength?

7. Line 249 I liked this analysis of SNPs. Perhaps also mention that synonymous SNPs are rare the ends of exons also and that this is associated with ESE disruption: Caceres, E. F., & Hurst, L. D. (2013). The evolution, impact and properties of exonic splice enhancers. *Genome Biol.* 14(12), R143, Article R143. <https://doi.org/10.1186/gb-2013-14-12-r143>; Carlini DB, Genut JE: Synonymous SNPs provide evidence for selective constraint on human exonic splicing enhancers. *J Mol Evol* 2006, 62:89–98.

8. Line 415 – isn't it striking that the one class of constraint not explained by UTH is the only one not significant – Radrizzani nominate miRNA binding sites as an exception as they are motif based so not general purpose devices. Worth noting that prior estimates suggested only a very low proportion of synonymous site conservation was attributable to miRNA pairing Hurst, L.D. (2006) Preliminary assessment of the impact of microRNA mediated regulation on coding sequence evolution in mammals. *Journal of Molecular Evolution* 63 174-182

9. Line 415 – same table. There are nuances that I am not sure are covered. The current authors ref 14 shows that selection can favour both conservation and avoidance of ESEs and RBP binding sites depending on context. ESE conservation is for example expected near exon intron borders (~ 70 nucleotides) but there should be avoidance in exon cores. Likewise, intron RBPs are avoided in exons and vice versa. Thus, at least for ESEs you might like to consider exonic sites near exon ends rather than all sites.

10. You also consider only RESCUE ESE as a set of putative ESEs – which is probably fine – the consensus INT3 set of Caceres and Hurst seems to behave the same way. However RESCUE contains only 238 motifs and so is unlikely to explain much. Expand the ESE set and you see more constraint (as with Savisaar and Hurst your ref 46).

11. Line 488 – this positive correlation between GC4 and GC intron is well described but the slope is far above 1 – ie the GC richest genes have a GC4 far above the neighbour intron. In part this is owing to TEs (rather AT rich) in the introns, but masking these doesn't remove the effect (Duret, L., & Hurst, L. D. (2001). The elevated GC content at exonic third sites is not evidence against neutralist models of isochore evolution. *Mol Biol Evol*, 18(5), 757-762. Radrizzani et al suggest this is consistent with their hypothesis – worth mentioning?

12.

Reviewer #2

(Remarks to the Author)

The authors present evidence for the unwanted transcript hypothesis that states that high GC content at 4d synonymous sites is the result of purifying selection, due to splice site conservation, regulatory elements conservation and because it is possible that high GC transcripts are distinguished from GC poor transcripts and as a result lead to greater export from the nucleus and translation. The evidence for purifying selection at 4d synonymous sites is based on the Zoonomia resource and the phyloP score which in turn is based on comparison of each site to freely mutating repeats in the genome. While conservation of splice site and regulatory elements is anticipated and is highly compatible with previous knowledge, the maintenance of high GC content through selection is debatable and requires stronger evidence.

Lines 262, 263 and Fig 4. Legend, "MAF" presumably stands for Minor Allele Frequency. This should be defined, at least at the first appearance, and possibly also in Fig. 4 legend for easier readability of the figure.

Line 262: " $4 \times 10^{-6} > \text{MAF} < 0.01$ " should be changed to " $4 \times 10^{-6} < \text{MAF} < 0.01$ ".

Line 263: " $0.01 > \text{MAF} < 0.5$ " should be changed to " $0.01 < \text{MAF} < 0.5$ ".

Fig. 4C - color legend is missing.

Line 484-487: "However, despite these differences which may be driven by gBGC, we demonstrate that ~21% of 4d sites are under significant evolutionary constraint across the 240 genomes, with a strong GC bias. This GC bias is unlikely to be explained by gBGC, as a GC bias would also be expected to be seen in the surrounding sequence and not specific to the 4d sites."

The authors claim that although mammalian GC content is likely associated to the degree of GC-biased gene conversion (gBGC) in a species they demonstrate that 21% of 4d sites are under significant evolutionary constraint, however the comparison in Zoonomia from which the PhyloP constraint score is derived is based on comparison to differences between mammals in freely mutating repeat sequences, which are non-genic and as such might be less prone to GC-biased gene conversion. Furthermore, in addition to gBGC also mismatch repair and transcription coupled repair were reported to be either GC-biased or biased toward the non-template strand. Transcription coupled repair is elevated in genes with high expression, therefore it is not obvious that the synonymous sites identified as 4d sites under evolutionary constraint, are not sites that experience more transcription coupled repair because they are sites in highly expressed genes. To support their claim the authors could perhaps show evidence for genes with many 4d constraint sites which are strongly regulated to have low expression. Alternatively, the authors should tone down their conclusions regarding the high percent of constraint 4d sites, and explain that while their results are compatible with UTH, they could also be explained by mechanistic GC-biases such as gBGC, mismatch repair and transcription coupled repair. Another possible interpretation of the data is that, while mechanistic GC-bias could be the cause of high GC4, this attribute of highly expressed genes is then utilized by the cell to identify those transcripts as important, suggesting that a mechanism for the identification of high GC transcript evolved after the bias towards high GC was present, and possibly the biased repair towards GC is maintained in evolution due to its importance in producing this distinction between more important and less important genes.

Line 488-490: "Whilst we do see a positive correlation between GC4 content of transcripts and local GC content, 4d site constraint in fact shows a slight negative correlation, with less 4d constraint observed in GC-rich regions."

This statement needs to be backed up by some statistical analysis, otherwise the data suggest that there is a significant positive correlation, and slight negative correlation might be based on the fitted curve, but at minimum a Pearson correlation above some 4d constraint score should be calculated to claim there is negative correlation there. Referencing the figure would also be helpful.

Line 498-500: "Our analysis of human genetic variation data revealed a strong bias in GC to AT mutations at 4d sites regardless of constraint which, in the absence of selection against them, should lead to a high AT content at 4d sites."

If the bias in GC to AT mutations at 4d sites does not differ between constrained and unconstrained sites, doesn't it imply that the level of purifying selection is comparable between constrained and unconstrained 4d sites? Otherwise, if constrained sites are under selection, while unconstrained sites are not under selection, wouldn't we expect to have a difference between the graphs of SNPs vs. MAF. If there is no difference then higher singleton SNPs at GC to AT could simply reflect mutational events that after a longer time tend to be less frequent due to genetic drift, and not due to selection. Adding a comparative figure equivalent to fig. 4D and 4E separated for constrained and unconstrained sites would be informative to distinguish between the two interpretations. Simply comparing the frequencies of mutation types between single allele and 0.01-0.05 suggest that GC to AT mutations have a similar proportion regardless whether it is a single allele SNP or it has an MAF of 0.01-0.05, suggesting that GC to AT mutations are not eliminated due to purifying selection (addition of pie charts of mutational types for each MAF category might be helpful to visualize if changes to proportions over different MAF categories exist, and a chi square test could be helpful in assessing whether the differences are significant).

Line 591-593: "As most genes contain multiple transcripts we chose an approach to select one representative transcript per protein coding gene to simplify analyses."

In the methods the authors state that only the one transcript has been chosen where alternative splice variants are documented. Therefore it is likely that many positions of alternative splice sites which are under constraint are not annotated as splice sites and as a result the constraint on 4d sites which are not explained due to splice sites conservation is overestimated. The best approach, in my opinion, would be to consider every possible splice site, in addition to the splice sites of the canonical transcript, and categorize splice sites into constitutive and alternative splice sites, to compare the level of constraint between them.

It would also be beneficial to distinguish or categorize 4d sites into highly and lowly expressed genes for each of the constrained 4d categories (splice sites, regulatory regions, etc.) to see how gene expression effects 4d constraint in each category.

Version 1:

Reviewer comments:

Reviewer #3

(Remarks to the Author)

Authors provide arguments for The unwanted transcript hypothesis (UTH) by analyzing a 240 placental mammal genome alignment and constraint scores. Their main argument is that a strong GC bias in constrained four-fold degenerate (4d) sites is evidence for the UTH.

As Reviewer 2, I find the rationale for excluding GC-biased gene conversion (gBGC) as an explanation for this pattern to be highly problematic.

Every pattern reported (80% of bases at constrained 4D sites are G or C, positive correlation between GC4 content and constraint) can be explained by gBGC. The arguments raised by the authors to exclude gBGC are not convincing.

The main argument of the authors is that it cannot be gBGC because "genes with high GC4 content show considerably higher GC content than that of their neighboring sequences." This is not a valid argument, as this pattern is universal in mammalian genomes (even in recombination hotspots known to be affected by gBGC) and has been known for over 20 years without dismissing the gBGC hypothesis. One hypothesis to explain this pattern is that higher sequence differences in non-coding regions reduce the likelihood of recombination, as heteroduplex DNA formation and propagation are more likely to occur in conserved coding and regulatory regions of the genome (see Birdsell et al. 2002, 10.1093/oxfordjournals.molbev.a004176). Additionally, it makes sense that coding regions, being more constrained, tend to be older and have undergone more recombination events than non-coding regions, which are typically younger and result from transposable elements that can be deleted without consequences following high recombination rates.

They also argue that "we observe a negative correlation between local GC content and transcript 4d site mean phyloP...showing that high 4D constraint is not associated with being in GC-rich genomic regions and therefore unlikely driven by gBGC." This is not convincing for dismissing gBGC. It is expected that recombination hotspots with extreme GC content are more likely to undergo GC->AT mutations (as mutation is AT-biased and GC is higher here) and frequently switch between AT and GC at 4d sites, particularly given the case that recombination hotspots are short-lived at a small scale but more conserved at a larger scale.

Minor Remarks

- *Sorex araneus* is expected to have extremely high N_e (as they are small and short-lived), but has the highest GC4. In Figure 5, N_e correlates negatively with GC4. How is it possible? Is it an exception?

- The statement "As expected if constrained sites are under purifying selection, a lower proportion for constrained sites are variable compared to unconstrained sites" seems circular. Constraint sites are expected to be less variable by definition.

Version 2:

Reviewer comments:

Reviewer #3

(Remarks to the Author)

The authors have addressed all my remarks and made significant modifications to their manuscript in a commendable manner. I am thoroughly satisfied with the changes made and require no more modifications.

J. Romiguier

Response to reviewers' comments

Reviewer #1

In this timely paper, the authors combine the recent zoonomia 240 species alignment with the even more recent unwanted transcript hypothesis (UTH). The latter is an attempt to resolve the enigma of commonplace selection on synonymous sites in mammals that have small effective population sizes and hence – according to the nearly neutral hypothesis – should have inefficient selection and hence an absence of selection on synonymous mutations. The UTH squares this circle by arguing that species with low N_e have in turn an acute problem with unwanted transcripts and so selection favours devices to reduce their rate of creation (HUSH, selection on splice control, 5' RNA pol II extension, intragene methylation to suppress intragenic transcription) or filter them out. Both, but especially the latter, can select for devices (export pathways, RNAses, P bodies etc) that favour GC rich transcripts (as mutation bias is GC->AT biased, neutral equilibrium is AT rich) this in turn favouring synonymous mutations that make a transcript appear more native.

The authors find abundant evidence that UTH makes a strong case for the constraint seen finding that constrained 4 fold sites tend to be GC rich, that intrapopulation variation suggests selection against GC reducing mutations, and that there is the predicted association with splice sites etc. The most striking result is that the trends run inverse to N_e as predicted by the model (an extension of Wu and Hurst's argument, as noted) and that recent TE insertions (the most likely to remain transcriptionally active?) predict the degree of constraint.

All in all the paper makes a strong case that the UTH is the framework within which to consider selection on synonymous sites in mammals. This then in broad terms is a valuable paper, not least because the common assumptions have been either a) that owing to low N_e there is no selection on synonymous sites in mammals (despite all the evidence to the contrary) or b) that codon usage bias is reflective of translational selection (seen in *E. coli*, yeast etc). I was also struck by the fact that gBGC appears not to explain the patterns.

In general, I have very few concerns about the quality of the analysis which appears to be very well done (and obviously to a significant scale). With all such analyses there are choices that could be different (I'll detail some of these). However, perhaps the biggest concern – although it isn't a big concern – is in the writing. I think it would be better to have some clear statements not just of results but to summarize the overall take. I was unsure in several places as to whether the authors considered the data in agreement with UTH or not. Indeed, the abstract could end with a definitive statement to the effect that "We conclude that the unwanted transcript hypothesis is the most viable model to explain the commonality of selection on synonymous sites in mammals" – or something to that effect.

RESPONSE:

We thank the reviewer for their positive and encouraging comments and that they appreciate the contribution this paper will bring to the field, particularly in terms of how the evidence we present strongly supports the predictions of the unwanted transcript hypothesis. We have now edited the text to include more clear statements about whether our data agrees with the UTH, as detailed in response to the reviewer's other comments below. We also now include a concluding line to the abstract to more clearly state that our findings provide strong support

towards the UTH (lines 26-28), as well as similar sentences in the discussion (lines 435-437).

Specific comments:

1. Specifically, in the section on CpG conservation, I wasn't clear what the authors think the relationship was with the UTH. The original UTH authors note that intragene body methylation is a strategy to reduce spurious intragene transcription (citing the current paper's ref 57, Neri et al). Neri et al indeed, note that intragene body methylation is there to suppress unwanted transcripts and is especially evident in the most highly expressed genes (those being in open chromatin much of the time we must presume and thus most vulnerable to spurious transcription). In this sense it makes perfect sense to me from the UTH point of view. However, there is also the problem that on the whole CpG is rare in human CDS so – according to UTH we have systems like ZAP and RIG-I to filter CpG rich transcripts. In this sense, at first sight, the UTH seems to want to have it both ways: it can explain why high CpG is filtered out but also why some genes might select for conserved CpG. It isn't however totally plastic as it suggests follow-on predictions: when CpG is conserved these should be in highly expressed genes (a la Neri et al) and, as Radrizzani suggest when discussing UpA rich genes, the RNA structure and other synonymous sites should evolve to prevent the CpGs from being exposed to ZAP etc. This leads to two predictions: genes with conserved CpG should be highly expressed and the other synonymous sites should be under selection to force RNA structure in a manner than conceals the CpGs. The first is readily tested – the second not so easy. But you could ask whether if there is an abundance of CpG's conserved do we also see evidence for conservation at other synonymous sites in the same gene – as suggested by by Schattner, P. & Diekhans, M. Regions of extreme synonymous codon selection in mammalian genes. *Nucleic Acids Res.* 34, 1700–1710 (2006) in the context of UpA. **This relates to my overall issue that the discussion could be stronger – it does seem to me that the UTH makes a remarkably strong case for what we see.**

RESPONSE:

We thank the reviewer for their interpretation and suggestions in this comment. We have followed this up by testing for the two things they suggest:

1) We used human gene expression data from Cardoso-Moreira et al. (2019), [Nature https://doi.org/10.1038/s41586-019-1338-5](https://doi.org/10.1038/s41586-019-1338-5) (data from 7 organs across multiple developmental time points) to show that genes enriched for CpG constraint are significantly more highly expressed than genes lacking CpG enrichment. This is the case in general and within each organ type. We also show that genes enriched for CpG constraint are active in more cells than other genes on average in the mouse single-cell organogenesis atlas, suggesting these genes are widely expressed, in agreement with the predictions of the UTH (lines 340-351).

2) We now show that genes enriched for 4d CpG constraint are also enriched for 4d constraint generally (after removing CpG sites). We state that there is a significant positive correlation between these two measures and that this is expected under the UTH (lines 336-340).

We have also edited the discussion throughout to present a stronger argument for how the evidence we present supports the UTH (lines 414 onwards).

2. Line 86, Transcripts that are AT rich....I think you might preface this saying “According to the UTH, transcripts... “Otherwise it isn’t clear whether this is your prediction or theirs.

RESPONSE:

We have added this suggested edit (line 79).

3. Line 149 – reflecting the hypermutability... according to UTH it could also be because we have filters like ZAP against CpG sites.

RESPONSE:

We now make reference to suppression by ZAP as suggested (lines 338-340).

4. Line 152 selection for epigenetic regulation – is regulation the only possibility? Surely as with Neri et al it could simply be suppression of intragene body spurious transcription?

RESPONSE:

We now also suggest that intragene body spurious transcription suppression may also lead to constraint on CpG sites and reference the Neri et al. 2017 paper (lines 315-318).

5. Line 212 and on – I couldn’t see that the hypothesis predicts a GC by exon length effect beyond avoidance of A3 in HUSH susceptible long exons (as evidenced in Radrizzani et al’s figure 3). Indeed, the original HUSH papers find the effect is not so much preference for GC as avoidance of A. For single exon genes I would expect a negative correlation as the GC preferences seem to be very much on the 5’ ends (Mordstein et al Cell Systems). Assuming the 5’ end necessary for this effect is a fixed size, the GC4 should be lower for longer transcripts. Mechanistically this may relate to the PolIII extension effect: Vlaming, H., Mimoso, C. A., Field, A. R., Martin, B. J. E. & Adelman, K. Screening thousands of transcribed coding and non-coding regions reveals sequence determinants of RNA polymerase II elongation potential. Nat. Struct. Mol. Biol. 29, 613–620 (2022) but we don’t yet know enough about RNA export and the role of the 5’ CDS end.

RESPONSE:

We have now edited this section to state that the negative correlation we show between GC4 content and exon length is predicted under the UTH due to the importance of 5` GC content. We reference the Mordstein paper for this (lines 172-175).

6. Line 234 – these splice junction proximal effects are likely to reflect selection on splice junctions more than GC content aren’t they – how do they relate to splice site strength?

RESPONSE:

We now state that the base biases at splice junctions likely reflect selection for effective splicing and cite two papers that show a bias for G at first and last exon positions of strong splice sites, in agreement with our data (lines 184-188).

7. Line 249 I liked this analysis of SNPs. Perhaps also mention that synonymous SNPs are rare the ends of exons also and that this is associated with ESE disruption: Caceres, E. F., & Hurst, L. D. (2013). The evolution, impact and properties of exonic splice enhancers. Genome Biol, 14(12), R143, Article R143. <https://doi.org/10.1186/gb-2013-14-12-r143>;

Carlini DB, Genut JE: Synonymous SNPs provide evidence for selective constraint on human exonic splicing enhancers. *J Mol Evol* 2006, 62:89–98.

RESPONSE:

We now include reference to this previous study in the discussion (line 470).

8. Line 415 – isn't it striking that the one class of constraint not explained by UTH is the only one not significant – Radrizzani nominate miRNA binding sites as an exception as they are motif based so not general purpose devices. Worth noting that prior estimates suggested only a very low proportion of synonymous site conservation was attributable to miRNA pairing Hurst, L.D. (2006) Preliminary assessment of the impact of microRNA mediated regulation on coding sequence evolution in mammals. *Journal of Molecular Evolution* 63 174-182

RESPONSE:

We now include a reference to this previous study that suggested only a small proportion of synonymous site constraint is due to miRNA binding (lines 360-363).

9. Line 415 – same table. There are nuances that I am not sure are covered. The current authors ref 14 shows that selection can favour both conservation and avoidance of ESEs and RBP binding sites depending on context. ESE conservation is for example expected near exon intron borders (~ 70 nucleotides) but there should be avoidance in exon cores. Likewise, intron RBPs are avoided in exons and vice versa. Thus, at least for ESEs you might like to consider exonic sites near exon ends rather than all sites.

RESPONSE:

We have now restricted our annotation of 4d sites within ESEs to those within 70 bp of exon-intron borders, as suggested by the reviewer. We show that constraint is significantly higher at 4d sites in ESEs within 70 bp of the borders compared to elsewhere in the exons (lines 638-641) and include this category in our glm of phyloP against annotations.

10. You also consider only RESCUE ESE as a set of putative ESEs – which is probably fine – the consensus INT3 set of Caceres and Hurst seems to behave the same way. However RESCUE contains only 238 motifs and so is unlikely to explain much. Expand the ESE set and you see more constraint (as with Savisaar and Hurst your ref 46).

RESPONSE:

In our analysis of ESEs we have considered both the RESCUE ESE set as well as the INT2 set of Caceres and Hurst (2013). This has now been updated in the methods (lines 633-638). We actually found that restricting our analysis to a more conservative set of ESEs explained more of the constraint observed at 4d sites. We reasoned that this may be due to evolution and turnover of ESEs among species, so a more conservative set may be more representative of ESEs present across mammals and hence higher constraint observed. Increasing our motif set to 520 by taking the INT3 set and all ESEs with a ESEseq score >0.5 from Ke et al (2011) actually explained less constraint, which may not be unexpected given that the Ke et al. set was shown to be fast evolving and contain a higher density of SNPs, so is unlikely to be highly conserved across mammals (Caceres and Hurst 2013).

We also looked to include the set of 2,069 putative exonic enhancers (PESE set) from Zhang and Chasin (2004), but their dataset is no longer available online and we have not had a reply from the authors after requesting it directly.

11. Line 488 – this positive correlation between GC4 and GC intron is well described but the slope is far above 1 – ie the GC richest genes have a GC4 far above the neighbour intron. In part this is owing to TEs (rather AT rich) in the introns, but masking these doesn't remove the effect (Duret, L., & Hurst, L. D. (2001). The elevated GC content at exonic third sites is not evidence against neutralist models of isochore evolution. *Mol Biol Evol*, 18(5), 757-762. Radrizzani et al suggest this is consistent with their hypothesis – worth mentioning?

RESPONSE:

We now provide the slope of the line in the results section to show it is greater than 1 - in fact, it is 2.17 (lines 221-223) - and expand on this point in the discussion to state that the GC4 content is well above the local GC content in genes with high GC4, and reference the Duret and Hurst paper to support the fact this isn't fully explained by TE content of the neighbouring sequence (lines 451-453).

Reviewer #2

1. The authors present evidence for the unwanted transcript hypothesis that states that high GC content at 4d synonymous sites is the result of purifying selection, due to splice site conservation, regulatory elements conservation and because it is possible that high GC transcripts are distinguished from GC poor transcripts and as a result lead to greater export from the nucleus and translation. The evidence for purifying selection at 4d synonymous sites is based on the Zoonomia resource and the phyloP score which in turn is based on comparison of each site to freely mutating repeats in the genome. While conservation of splice site and regulatory elements is anticipated and is highly compatible with previous knowledge, the maintenance of high GC content through selection is debatable and requires stronger evidence.

RESPONSE:

We thank the reviewer for their fair and helpful comments. Following other comments from the reviewer regarding the strength of evidence for selection for high GC content at 4d sites, we now present a more cohesive argument as to how our results are suggestive of selection as the most likely explanation for the observed strong GC bias at constrained synonymous sites. In particular, the observations that GC content of constrained 4d sites is substantially greater than that observed in neighbouring sequence, that sites are so highly conserved across so many genomes, and that constrained sites are lacking genetic variation in human populations are all unlikely to result from mechanistically generated biases such as gBGC, mismatch repair, and transcription-coupled repair. These arguments are laid out in the discussion (lines 439-470).

2. Lines 262, 263 and Fig 4. Legend, "MAF" presumably stands for Minor Allele Frequency. This should be defined, at least at the first appearance, and possibly also in Fig. 4 legend for easier readability of the figure.

RESPONSE:

This has now been defined in the text (line 198) and figure 4 legend

3. Line 262: " $4 \times 10^{-6} > \text{MAF} < 0.01$ " should be changed to " $4 \times 10^{-6} < \text{MAF} < 0.01$ ".

RESPONSE:

The text in this section has been edited and no longer contains this statement

4. Line 263: " $0.01 > \text{MAF} < 0.5$ " should be changed to " $0.01 < \text{MAF} < 0.5$ "

RESPONSE:

The text in this section has been edited and no longer contains this statement

5. Fig. 4C - color legend is missing.

RESPONSE:

The colour legend in figure 4 applies to all the plots, but in C only the transitions are shown. This is now clarified in the legend text.

6. Line 484-487: "However, despite these differences which may be driven by gBGC, we demonstrate that ~21% of 4d sites are under significant evolutionary constraint across the 240 genomes, with a strong GC bias. This GC bias is unlikely to be explained by gBGC, as a GC bias would also be expected to be seen in the surrounding sequence and not specific to the 4d sites."

The authors claim that although mammalian GC content is likely associated to the degree of GC-biased gene conversion (gBGC) in a species they demonstrate that 21% of 4d sites are under significant evolutionary constraint, however the comparison in Zoonomia from which the PhyloP constraint score is derived is based on comparison to differences between mammals in freely mutating repeat sequences, which are non-genic and as such might be less prone to GC-biased gene conversion. Furthermore, in addition to gBGC also mismatch repair and transcription coupled repair were reported to be either GC-biased or biased toward the non-template strand. Transcription coupled repair is elevated in genes with high expression, therefore it is not obvious that the synonymous sites identified as 4d sites under evolutionary constraint, are not sites that experience more transcription coupled repair because they are sites in highly expressed genes. To support their claim the authors could perhaps show evidence for genes with many 4d constraint sites which are strongly regulated to have low expression. Alternatively, the authors should tone down their conclusions regarding the high percent of constraint 4d sites, and explain that while their results are compatible with UTH, they could also be explained by mechanistic GC-biases such as gBGC, mismatch repair and transcription coupled repair. Another possible interpretation of the data is that, while mechanistic GC-bias could be the cause of high GC4, this attribute of highly expressed genes is then utilized by the cell to identify those transcripts as important, suggesting that a mechanism for the identification of high GC transcript evolved after the bias towards high GC was present, and possibly the biased repair towards GC is maintained in evolution due to its importance in producing this distinction between more important and less important genes.

RESPONSE:

We thank the reviewer for this detailed comment and understand their concerns regarding the strength of the evidence for selection leading to high GC content at synonymous sites in mammals. The ancestral repeats that act as the neutral model when calculating phyloP are sampled from across the genome and we do not see any reason to suspect there would be a bias in the impact of gBGC on these repeats compared to coding regions. The effects of gBGC are seen at sites of recombination and, as recombination hotspots are continually evolving, a large proportion of the genome will be affected by gBGC over time with no bias towards coding sequence (Glémin et al, 2015, 10.1101/gr.185488.114). Additionally, if gBGC was the driver of conservation at synonymous sites in mammals we would expect to see that our constraint scores correlate with local GC content, which they do not. In fact, we see a distinct lack of constraint in regions of high or low GC content and the GC content of 4d sites is often much higher than in the neighbouring sequence (now stated on lines 451-453 after reviewer 1's comment 11). With regards to transcription coupled repair, it is not clear that this is likely to cause a GC-bias, but we now show that many genes enriched for constraint at synonymous site constraint are very lowly expressed, suggesting this is unlikely to explain the patterns we see (lines 215-238). Also, GC4 content appears independent from expression level, regardless of constraint. If transcription coupled repair led to higher GC4 content in more highly expressed genes then GC4 content and expression level should correlate, but they do not.

7. Line 488-490: "Whilst we do see a positive correlation between GC4 content of transcripts and local GC content, 4d site constraint in fact shows a slight negative correlation, with less 4d constraint observed in GC-rich regions."

This statement needs to be backed up by some statistical analysis, otherwise the data suggest that there is a significant positive correlation, and slight negative correlation might be based on the fitted curve, but at minimum a Pearson correlation above some 4d constraint score should be calculated to claim there is negative correlation there. Referencing the figure would also be helpful.

RESPONSE:

This is in the discussion so we do not give the statistics again here. They are provided in the results section (lines 218-226) and shown in figure S1. This section has also been edited slightly in light of reviewer 1's comment 11.

8. Line 498-500: "Our analysis of human genetic variation data revealed a strong bias in GC to AT mutations at 4d sites regardless of constraint which, in the absence of selection against them, should lead to a high AT content at 4d sites."

If the bias in GC to AT mutations at 4d sites does not differ between constrained and unconstrained sites, doesn't it imply that the level of purifying selection is comparable between constrained and unconstrained 4d sites? Otherwise, if constrained sites are under selection, while unconstrained sites are not under selection, wouldn't we expect to have a difference between the graphs of SNPs vs. MAF. If there is no difference then higher singleton SNPs at GC to AT could simply reflect mutational events that after a longer time tend to be less frequent due to genetic drift, and not due to selection. Adding a comparative

figure equivalent to fig. 4D and 4E separated for constrained and unconstrained sites would be informative to distinguish between the two interpretations. Simply comparing the frequencies of mutation types between single allele and 0.01-0.05 suggest that GC to AT mutations have a similar proportion regardless whether it is a single allele SNP or it has an MAF of 0.01-0.05, suggesting that GC to AT mutations are not eliminated due to purifying selection (addition of pie charts of mutational types for each MAF category might be helpful to visualize if changes to proportions over different MAF categories exist, and a chi square test could be helpful in assessing whether the differences are significant).

RESPONSE:

We do see differences in the frequency of GC to AT alleles between constrained and unconstrained sites. Whilst singleton GC to AT SNPs are the most common type of SNP at both constrained and unconstrained sites (figures 4 a and b), this does not imply that the selection pressures are equal - these are somatic SNPs and their abundance can be explained by the GC to AT mutation bias. However, we do observe differences in the allele frequencies of constrained and unconstrained 4d sites, and GC to AT variants are rarer at sites under high constraint. We now report statistics to show that a smaller proportion of constrained sites contain SNPs in the TOPMed dataset compared to unconstrained sites and allele frequencies are lower at constrained sites, as expected if constrained sites are under purifying selection (lines 196-200). We also show that there is a significant negative correlation between phyloP and allele frequencies of GC to AT SNPs, whereas there is no relationship between phyloP and AT to GC SNP allele frequencies (lines 207-213). We have edited figure 4, removing panels D and E and replacing them with a single panel D showing SNP proportions for each mutation type against minor allele frequency for constrained and unconstrained sites separately.

9. Line 591-593: "As most genes contain multiple transcripts we chose an approach to select one representative transcript per protein coding gene to simplify analyses."

In the methods the authors state that only the one transcript has been chosen where alternative splice variants are documented. Therefore it is likely that many positions of alternative splice sites which are under constraint are not annotated as splice sites and as a result the constraint on 4d sites which are not explained due to splice sites conservation is overestimated. The best approach, in my opinion, would be to consider every possible splice site, in addition to the splice sites of the canonical transcript, and categorize splice sites into constitutive and alternative splice sites, to compare the level of constraint between them.

RESPONSE:

We appreciate that by taking only one transcript we are not characterising all the constraint at 4d sites due to splicing. We have now identified all human splice sites from the human gencode annotation and intersected it with the 4d sites in our analysis. This has revealed a further 5,612 'alternative' splice sites. However, this set of sites has a much lower mean phyloP (0.031) than the 'constitutive' set (3.55), suggesting that most of them are not under constraint in mammals. We do see that 1,267 of these 'alternative' splice sites are under constraint at the phyloP threshold of 2.27 (0.23% of all constrained 4d sites). We have now added a new line to table one to present the constraint observed in these alternative splice sites, as well as updates to the text (lines 189-192).

10. It would also be beneficial to distinguish or categorize 4d sites into highly and lowly expressed genes for each of the constrained 4d categories (splice sites, regulatory regions, etc.) to see how gene expression effects 4d constraint in each category.

RESPONSE:

We have now used the gene expression data from Cardoso-Modeira et al 2019 to categorise genes into highly (80th percentile), intermediate (20th-80th percentiles) and lowly (20th percentile of expression) expressed genes. Re-running our linear model of phyloP against constraint categories for highly and lowly expressed genes independently shows very little difference from the main model, i.e. the constraint explained in the models does not appear to relate to expression levels. To this end, we have now performed a linear model of gene expression against regulatory feature overlap for constrained 4d sites (lines 373-383) and made a new supplementary figure showing that 4d site overlap of various regulatory features does not vary much among expression levels (Figure S7). Whilst the model is significant (i.e. there are differences in the overlap of constrained 4d sites with various constraining features that relates to gene expression), it has an r^2 of 0.038 and so explains very little of the variance.

The feature with the greatest relative importance in the model of 0.98, capturing nearly all the explained variance, is overlap with RNA binding protein binding sites. Here, 4d sites in lowly expressed genes show significantly less overlap with RBP binding sites than more highly expressed genes. This is independent from constraint, i.e. overlap with RBP binding sites explains very little of the constraint observed at 4d sites (as shown in our main model) and the difference likely reflects the level of control of transcripts by RBPs depending on expression levels. We now detail this in the text (lines 373-383).

Response to reviewer 3's comments

Our responses to the reviewer's comments are outlined in blue italicised text after each comment.

Reviewer #3 (Remarks to the Author):

Authors provide arguments for The unwanted transcript hypothesis (UTH) by analyzing a 240 placental mammal genome alignment and constraint scores. Their main argument is that a strong GC bias in constrained four-fold degenerate (4d) sites is evidence for the UTH.

As Reviewer 2, I find the rationale for excluding GC-biased gene conversion (gBGC) as an explanation for this pattern to be highly problematic.

Every pattern reported (80% of bases at constrained 4D sites are G or C, positive correlation between GC4 content and constraint) can be explained by gBGC. The arguments raised by the authors to exclude gBGC are not convincing.

We thank the reviewer for their comments and agree that we did not show a comprehensive consideration for the impact of gBGC on synonymous site constraint. We have now made substantial changes to the manuscript to include greater discussion of gBGC, which we detail below, and have included reference to more of the literature that demonstrates its likely impact on the GC content of mammalian genomes. This has helped to reframe the way we discuss our findings and show how some of our results can be explained by gBGC, without having to invoke selection. To this end, we have decided to swap the use of 'constraint' when describing conserved sites as this implies purifying selection. Instead, we now use 'conserved' and 'conservation', as sequence conservation does not imply selection and leaves open the possibility that selectively neutral processes such as gBGC can also lead to the high sequence conservation we observe.

*We have revised the abstract, introduction, and discussion (highlighted in blue) to include greater consideration of the impacts of gBGC, distinguishing more clearly between patterns that can be explained by gBGC and those that cannot. In addition, we have included a new set of analyses that demonstrate that some of the conservation observed at 4d sites likely relates to mutation rate heterogeneity within and among genes. We present these new findings in the results section (lines 373-436), in the discussion (lines 595-611), and we have **updated figure 6** to include presentation of these new results.*

The main argument of the authors is that it cannot be gBGC because "genes with high GC4 content show considerably higher GC content than that of their neighboring sequences." This is not a valid argument, as this pattern is universal in mammalian genomes (even in recombination hotspots known to be affected by gBGC) and has been known for over 20 years without dismissing the gBGC hypothesis. One hypothesis to explain this pattern is that higher sequence differences in non-coding regions reduce the likelihood of recombination, as heteroduplex DNA formation and propagation are more likely to occur in conserved coding and regulatory regions of the genome (see Birdsell et al. 2002, 10.1093/oxfordjournals.molbev.a004176). Additionally, it makes sense that coding regions, being more constrained, tend to be older and have undergone more recombination events

than non-coding regions, which are typically younger and result from transposable elements that can be deleted without consequences following high recombination rates.

This text resulted from a response to a comment from reviewer 1, where they suggested that such high GC content compared to neighbouring sequences would not be expected under a model of gBGC. We have now altered the text to acknowledge that differences between GC4 and neighbouring GC content are expected under gBGC due to the impact of TE insertions on non-coding sequence and the higher likelihood of recombination (and therefore gBGC) affecting constrained, coding sequences (lines 520-528). We now also include reference to the Birdsell paper.

*We have updated our analysis of the human SNP data to better assess the potential effect of gBGC on the conservation we observe at 4d sites. We show that there is a general negative correlation between phyloP and amount of variation at 4d sites, suggesting greater purifying selection at sites with higher phyloP overall. However, we also show that there is an excess of GC→AT variants at conserved 4d sites, which is not expected if the sites are under purifying selection but is expected if due to gBGC. This same analysis also revealed a deficiency in AT→GC variants at conserved 4d sites, which is suggestive of selection and cannot be explained by gBGC (lines 216-231 in results section, and 563-573 in the discussion and a **new figure 4** to represent these results).*

They also argue that "we observe a negative correlation between local GC content and transcript 4d site mean phyloP...showing that high 4D constraint is not associated with being in GC-rich genomic regions and therefore unlikely driven by gBGC." This is not convincing for dismissing gBGC. It is expected that recombination hotspots with extreme GC content are more likely to undergo GC->AT mutations (as mutation is AT-biased and GC is higher here) and frequently switch between AT and GC at 4d sites, particularly given the case that recombination hotspots are short-lived at a small scale but more conserved at a larger scale.

*We have now revised the text so as to not suggest that the negative correlation between local GC content and transcript 4d site mean phyloP is evidence against gBGC. We suggest it shows that 4d conservation is lower in regions more recently affected by gBGC and this likely relates to rapidly evolving recombination landscapes (lines 489-494). Additionally, we acknowledge that more ancient genes will have experienced more gBGC over their history, likely leading to higher GC content. Whilst this may be true for some genes, we show that GC4 content is not a good predictor of 4d site conservation and genes with highly conserved 4d sites do not necessarily have high GC4 content (and many do not!), suggesting that the conservation is not due to gBGC. In fact, we see a strong positive correlation between phyloP at GC 4d sites and AT 4d sites within transcripts, suggesting that high 4d site conservation in certain genes is independent from gBGC (lines 538-546). We have replaced **figure 1G** with a plot that shows this correlation between phyloP at GC and AT sites within transcripts.*

Minor Remarks

- *Sorex araneus* is expected to have extremely high N_e (as they are small and short-lived), but has the highest GC4. In Figure 5, N_e correlates negatively with GC4. How is it possible? Is it an exception?

Figure 5 depicts GC4 content of the set of 4d sites present in all 240 species, rather than the complete data set where species coverage was not required to be 100%. Whilst GC4 at this subset of 4d sites negatively correlates with N_e , we do not see a significant correlation between GC4 and N_e when considering all 4d sites. In some lineages we do see significant positive correlations between N_e and total GC4 proportion, as expected under a model of gBGC. We have now added this to the text (lines 279-283 plus new supplementary figure S3). We did not have N_e estimates for *S. araneus* but, as you suggest, it likely has an extremely high N_e and this fits with its high genome-wide GC4 content. Yet its GC4 content in the subset of 4d sites present in all 240 genomes is in the lower third of species (although still high at 0.74), fitting with the general pattern of a negative correlation between N_e and GC4 at these sites.

- The statement "As expected if constrained sites are under purifying selection, a lower proportion for constrained sites are variable compared to unconstrained sites" seems circular. Constraint sites are expected to be less variable by definition.

We agree that constrained sites are expected to be less variable by definition and believe our switch to using 'conserved' rather than 'constrained' throughout the manuscript addresses this - a site conserved across mammals **may** be under constraint/purifying selection and this can be evidenced by a lack of variation at the site, which we show. We have now edited the text to remove this circularity (lines 216-231).